# N6-methyladenosine in DNA promotes genome stability

Brooke A Conti[1]*, Leo Novikov[1], Deyan Tong[2], Qing Xiang[2], Savon Vigil[3], Thomas J McLellan[3], Chuong Nguyen[3], Nancy De La Cruz[4], Reshma T Veettil[4], Prashant Pradhan[4], Parag Sahasrabudhe[3], Jason D Arroyo[5], Lei Shang[5], Benjamin R Sabari[4], David J Shields[1], Mariano Oppikofer[1]*

[1]Centers for Therapeutic Innovation, Emerging Sciences and Innovation, Pfizer, New York, United States; [2]Target Sciences, Emerging Sciences and Innovation, Pfizer, New York, United States; [3]Discovery Sciences, Pfizer, Groton, United States; [4]Laboratory of Nuclear Organization, Cecil H. and Ida Green Center for Reproductive Biology Sciences, Division of Basic Research, Department of Obstetrics and Gynecology, Department of Molecular Biology, Hamon Center for Regenerative Science and Medicine, University of Texas Southwestern Medical Center, Dallas, United States; [5]Target Sciences, Emerging Sciences and Innovation, Pfizer, Cambridge, United States

**\*For correspondence:**
bconti830@gmail.com (BAC);
mariano.oppikofer@gmail.com
(MO)

## eLife Assessment

This manuscript reports **important** findings that the methyltransferase METTL3 is involved in the repair of abasic sites and uracil in DNA, mediating resistance to floxuridine-driven cytotoxicity. The presented evidence is conclusive for the involvement of m6A in DNA involving single cell imaging and mass spectrometry data. The authors present **convincing** evidence that the m6A signal does not result from bacterial contamination or RNA.

**Abstract** DNA base lesions, such as incorporation of uracil into DNA or base mismatches, can be mutagenic and toxic to replicating cells. To discover factors in repair of genomic uracil, we performed a CRISPR knockout screen in the presence of floxuridine, a chemotherapeutic agent that incorporates uracil and fluorouracil into DNA. We identified known factors, such as uracil DNA N-glycosylase (UNG), and unknown factors, such as the N6-adenosine methyltransferase, METTL3, as required to overcome floxuridine-driven cytotoxicity. Visualized with immunofluorescence, the product of METTL3 activity, N6-methyladenosine, formed nuclear foci in cells treated with floxuridine. The observed N6-methyladenosine was embedded in DNA, called 6mA, and these results were confirmed using an orthogonal approach, liquid chromatography coupled to tandem mass spectrometry. METTL3 and 6mA were required for repair of lesions driven by additional base-damaging agents, including raltitrexed, gemcitabine, and hydroxyurea. Our results establish a role for METTL3 and 6mA in promoting genome stability in mammalian cells, especially in response to base damage.

## Introduction

DNA base lesions, including incorporation of uracil into DNA and base mismatches, can be mutagenic and toxic to replicating cells (reviewed in *Thompson and Cortez, 2020*). Uracil can be incorporated into the genome through several means, including DNA replication, cytosine deamination, and

**eLife digest** All mammals store their genetic material in the form of DNA, which is constantly damaged by factors such as ultraviolet radiation, chemicals, and errors during cellular processes. To prevent such damage from causing harmful mutations, it is important that cells have repair mechanisms that can fix damaged DNA.

Some drugs used to treat cancer cause damage to DNA by incorporating uracil, a compound that doesn't belong in DNA. This can lead to DNA mutations if not repaired. An enzyme known as UNG2 is involved in repairing this damage by removing the uracil-based lesions. However, the process of uracil repair was not fully understood.

To investigate, Conti et al. treated cancer cells with the drug floxuridine, which is known to cause uracil-based DNA damage. A genetic screening technique identified that a gene encoding an enzyme known as METTL3 is required for repairing uracil-related damage. Further experiments suggested that METTL3 adds markers known as m6A to DNA to help direct repair by UNG2. Inhibiting METTL3 made the cells more sensitive to the drug treatment and reduced the amount of UNG2 at sites of DNA damage.

While m6A marks are known to exist in bacterial DNA, evidence of them in mammalian DNA has been a topic of debate. The findings of Conti et al. suggest that these modifications form in response to DNA damage and help to facilitate repair DNA in mammalian cells. Further research is needed to clarify how METTL3 and m6A marks interact with other DNA repair pathways. Gaining a greater understanding of these repair processes could help future research into strategies to treat diseases driven by DNA damage, such as cancer.

exposure to FDA-approved chemotherapeutic agents, such as fluorouracil (FU) and floxuridine, which can introduce uracil (U) and FU into the genome as U:G, U:A, FU:G, or FU:A pairs (*Christenson et al., 2021*; *Meyers et al., 2005*). These drugs are commonly used to treat solid tumors, such as colorectal cancer (CRC).

Removal of uracil and downstream DNA repair are necessary to maintain genome integrity. Genomic uracil is removed through two molecular pathways: uracil base excision repair (U-BER) and mismatch repair (MMR) (*Christenson et al., 2021*; *Meyers et al., 2005*; reviewed in *Krokan and Bjørås, 2013*). U-BER plays a predominant role, and it depends on uracil DNA N-glycosylase (UNG), a highly conserved enzyme that cleaves the N-glycosylic bond between a uracil or FU base and the DNA backbone, whether in U:A, U:G, FU:A, or FU:G pairs. UNG is expressed as two isoforms with different N-terminal regions, UNG1 and UNG2. The N-terminal extension of UNG2 contains residues required for nuclear localization (*Otterlei et al., 1998*) and is the isoform of interest in the work presented here. MMR plays a secondary role and depends on MutSα, a heterodimer composed of Mut S Homolog 2 (MSH2) and Mut S Homolog 6 (MSH6), which recognizes single-nucleotide mismatches in DNA relevant to uracil removal (U:G and FU:G). MutSα can also recognizes U:A and FU:A pairs, albeit with low efficiency (*Fischer et al., 2007*; *Meyers et al., 2005*). Following initial damage recognition by UNG or MutSα, many additional factors cooperate to return the genome to its original state (reviewed in *Jiricny, 2013*; *Krokan and Bjørås, 2013*).

Understanding the molecular mechanisms of uracil repair is therapeutically relevant. For example, U-BER and MMR pathways can counteract the efficacy of chemotherapy (*Christenson et al., 2021*), and thus these pathways may contain targets for clinical applications (reviewed in *Gohil et al., 2023*). Additionally, the coordination of uracil repair in response to programmed cytosine deamination in B cells is vital for normal antibody maturation, and alterations in these repair pathways can lead to cancer development or immunodeficiencies (reviewed in *Kavli et al., 2007*). In this study, we took an unbiased, functional genomics approach to find novel factors involved in uracil repair with clinical interest, which identified factors involved in methylation of adenosine at the N6 position, such as methyltransferase-like 3 (METTL3) and methyltransferase-like 14 (METTL14).

The N6-adenosine methyltransferase complex is composed of METTL3, METTL14, and Wilms tumor 1-associated protein (WTAP). METTL3 contains a functional methyltransferase domain belonging to the MT-A70 family of S-adenosyl-methionine-dependent methyltransferases and is well studied for its ability to methylate adenosine at the N6 position in RNA (*Liu et al., 2014*). The literature suggests that

METTL3 functions in DNA damage responses, with METTL3-dependent methylation of adenosine on RNA occurring in response to damage induced by UV and 5-FU (*Li et al., 2022*; *Xiang et al., 2017*). Moreover, METTL3 has been shown to interact with MMR factors, MSH2 and MSH6 (*Yue et al., 2018*) while functional analyses reveal that METTL3 is involved in MMR (*Zhou et al., 2022*). To date, these results have primarily been interpreted based on METTL3's well-known ability to modify RNA and modulate mRNA stability in an N6-adenosine methyltransferase-dependent manner (reviewed in *Jiang et al., 2021*).

In eukaryotes, N6-methyladenosine has been well studied in RNA and m6A is commonly used when referring to the RNA species. However, the presence and function of the equivalent modification in DNA, referred to as 6mA, are less understood. In 2015, 6mA was first identified in DNA in *Caenorhabditis elegans* and *Drosophila* (*Greer et al., 2015*; *Zhang et al., 2015*) and the identification of 6mA in mammalian cells followed shortly in 2016 (*Wu et al., 2016*). Interestingly, using ChIP-seq, *Wu et al., 2016* found that genomic DNA (gDNA) 6mA is enriched at H2A.X sites. However, since the identification of 6mA in mammalian cells, the field has been plagued with challenges that call into question whether 6mA truly exists or is an artifact (reviewed in *Feng and He, 2023*). These challenges include the extremely low abundance of 6mA, prevalent contamination from bacterial DNA as a source of 6mA, and a lack of functional relevance for the 6mA modification in eukaryotic cells (reviewed in *Feng and He, 2023*).

In this study, we provide evidence that 6mA is greatly enriched in the DNA of mammalian cells in response to DNA damage and is required for DNA repair. The enrichment of m6A is specific for DNA and not RNA, as determined by orthogonal methods, and depends on METTL3's catalytic activity. Consistent with the functional genomics data presented, discovery proteomics revealed that components of the N6-adenosine methyltransferase complex associate with the DNA repair factor UNG2. We show that METTL3 and 6mA facilitate repair of damage caused by uracil-based chemotherapeutic agents, functioning upstream of UNG2 in U-BER. Additionally, we establish a broader role for METTL3

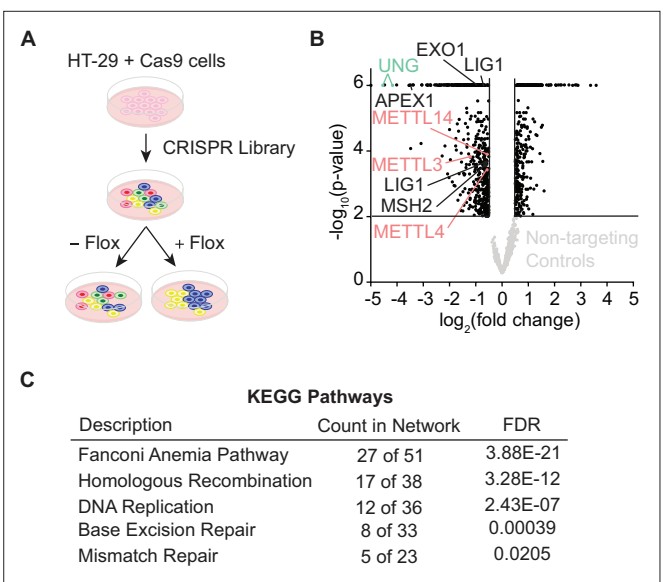

**Figure 1.** Whole-genome CRISPR screen identifies N6-methyltransferases in repair of floxuridine-induced DNA lesions. (**A**) Schematic of whole-genome CRISPR screen in HT-29 cells reported in B. (**B**) Volcano plot displaying MAGeCK gene level $\log_2$(fold change) for each gene in treated and untreated arms versus $-\log_{10}$(p-value). Cut-off displays genes with $\log_2$(fold change) >|0.5| and $-\log_{10}$(p-value) >2. Genes whose loss sensitizes cells to floxuridine skew to the left. Non-targeting guides are shown in light gray and fall below cut-off values. Essential genes also performed as expected, dropping out at later time points (data not shown). (**C**) KEGG pathway analysis for genes that sensitize cells to floxuridine with $\log_2$(fold change) >|0.5| and $-\log_{10}$(p-value) >2. Flox, floxuridine; FDR, false discovery rate.

The online version of this article includes the following source data and figure supplement(s) for figure 1:

**Figure supplement 1.** Generation of mCherry-tagged UNG KO DLD-1 cells.

**Figure supplement 1—source data 1.** Source data for panel B.

and 6mA in DNA repair, specifically in responding to base damage beyond uracil incorporation. This is the first evidence of a mechanistic link between 6mA deposition in DNA and DNA repair in mammalian cells.

## Results

### N6-adenosine-methyltransferases function in repair of floxuridine-induced DNA lesions

To identify modulators of response to floxuridine-induced DNA lesions, we performed a whole-genome Clustered Regularly Interspaced Short Palindromic Repeats (CRISPR) knockout screen in HT-29 cells, a MMR-proficient CRC cell line (*Figure 1A, B, Supplementary file 1*). UNG, targeted with guides that cut both isoforms, was identified as a top hit validating that the screen was successful (*Figure 1A, B*). Targeted genes sensitizing cells to floxuridine were associated with DNA repair and replication pathways (*Figure 1C*). Known factors in the repair of uracil lesions, including downstream repair factors in U-BER, such as apurinic/apyrimidinic endodeoxyribonuclease 1 (APEX1) and Ligase 1 (LIG1), and MMR factors, such as MSH2 and exonuclease 1 (EXO1), were among those identified as hits (*Figure 1A–C*). Interestingly, we also observed that the loss of METTL3, METTL14, and methyltransferase-like 4 (METTL4), sensitized cells to floxuridine (*Figure 1A, B*).

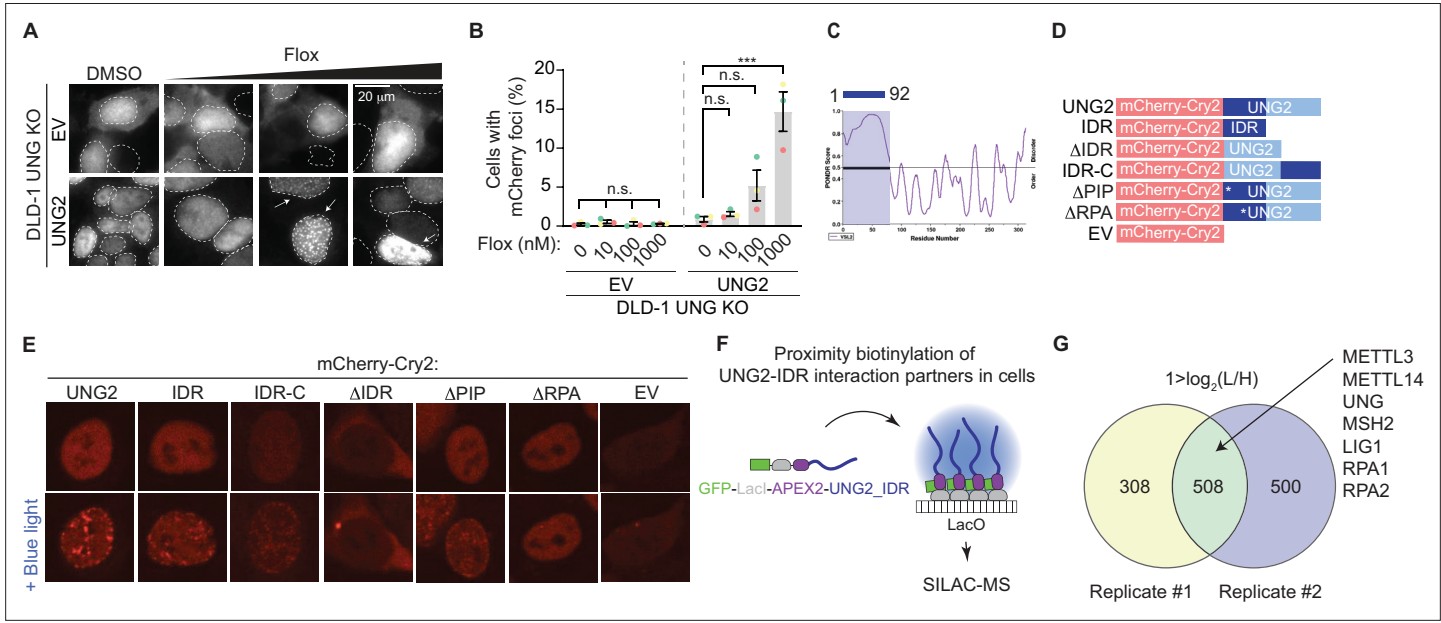

**Figure 2.** Discovery-based proteomics identifies N6-methyltransferases at sites of UNG2-seeded condensates. (**A**) Representative images from DLD-1 UNG KO cells expressing indicated mCherry-tagged cDNAs upon treatment with increasing concentrations of floxuridine at 64 hr post-treatment. (**B**) Quantification of experiment represented in A for percentage of cells with >5 mCherry foci. Error bars, mean ± SEM; ordinary one-way ANOVA with Dunnett's multiple comparisons test with a single pooled variance, ***p ≤ 0.001, n = 3 biological replicates. Statistical tests performed within individual groups, EV or UNG2, respectively. (**C**) PONDR VSL2 plot of disorder for UNG2. (**D**) Representative schematic of mutant UNG2 cDNA constructs expressed in UNG KO DLD-1 cells in E. IDR-only cDNA lacks amino acids 93–313, ΔIDR cDNA lacks amino acids 1–92, and IDR-C cDNA moved amino acids 1–92 to the C-terminus. (**E**) Representative images from DLD-1 UNG KO cells expressing indicated mCherry-Cry2-tagged cDNAs without or with stimulation of blue light for 60 s. While ΔIDR and EV images display cytoplasmic foci, these lack the distinct nuclear foci patterning observed for UNG2, IDR, and IDR-C constructs. (**F**) Schematic of proximity biotinylation of IDR interaction partners in cells. (**G**) Venn diagrams of factors identified by stable isotope labeling of amino acids in cell culture (SILAC)-based mass spectrometry (MS) with 1 > log$_2$(L/H), reflecting a fourfold enrichment in UNG-IDR/Control, from two biological replicates. n.s., non-statistically significant; Flox, floxuridine; EV, empty vector; IDR, intrinsically disordered region.

The online version of this article includes the following source data and figure supplement(s) for figure 2:

**Source data 1.** Source data for panel B.

**Figure supplement 1.** UNG2 responds to uracil-based DNA damage in a manner partly dependent on its intrinsically disordered region.

**Figure supplement 1—source data 1.** Source data for panels A, C, E, G and H.

In a complementary investigation into repair of uracil-based DNA lesions, we immunoprecipitated endogenous UNG2 and identified co-purifying factors by liquid chromatography coupled to tandem mass spectrometry (LC–MS/MS). Experiments were performed in the absence or presence of floxuridine to establish networks of factors that interact with UNG2 at baseline or under DNA damage conditions, respectively. Coimmunoprecipitations of UNG2 identified known BER factors, such as LIG1, but also WTAP, a key member of the N6-adenosine methyltransferase complex, which was identified in both conditions (*Supplementary file 2*). These data reinforce the notion that N6-methyladenosine-depositing enzymes are relevant for UNG2 activity.

While UNG2 is well studied for its enzymatic role in repair of uracil-containing DNA, the dynamics of UNG2's cellular localization in response to uracil-based DNA damage are incompletely understood. We found that UNG2 forms nuclear foci upon treatment with floxuridine (*Figure 2A, B*). In these experiments, a UNG2 cDNA construct was tagged with mCherry and expressed in UNG knockout (KO) DLD-1 cells, where both isoforms of UNG were targeted (*Figure 1—figure supplement 1A–C*, *Figure 1—figure supplement 1—source data 1*). Both wild-type DLD-1 and UNG KO cells exhibit about 30% of cells in S phase at baseline (*Figure 2—figure supplement 1A*). Using this system, we observed a significant increase in the percentage of cells with mCherry (UNG2) foci in response to uracil-based DNA damage. When cells were treated with increasing concentrations of floxuridine, nuclear foci were visualized in UNG2-expressing cells, but not in cells expressing mCherry alone (EV) (*Figure 2A, B*). Comparable results were observed upon treatment with raltitrexed, an inhibitor of thymidylate synthase that also increases genomic uracil (*Figure 2—figure supplement 1B, C*, *Figure 1—figure supplement 1—source data 1*). Additionally, the percentage of cells with UNG2 foci also increased upon treatment with hydroxyurea (HU), a ribonucleotide reductase inhibitor that alters nucleotide pools, but not gemcitabine, a cytosine analog, or mitomycin C (MMC), an interstrand crosslinker (*Figure 2—figure supplement 1D, E*). To assess whether DNA damage levels were linked to the ability of cells to form UNG2 foci, we quantified the mean nuclear γH2AX intensity after each treatment. The levels of DNA damage, as measured by mean nuclear γH2AX staining, did not correlate with the percentage of cells displaying UNG2 foci (*Figure 2—figure supplement 1F, G*). Together, these data suggest that UNG2 foci form specifically in response to uracil-based DNA damage.

Biomolecular condensates have emerged as an important organization principle inside cells and regulate many cell processes, including DNA repair (reviewed in *Banani et al., 2017*; *Conti and Oppikofer, 2022*; *Sabari et al., 2020*; *Shin and Brangwynne, 2017*). Thus, we hypothesized that uracil-induced UNG2 foci may represent functional condensates that concentrate factors required for uracil removal and DNA repair. Condensate formation is often driven by intrinsically disordered regions (IDRs) within proteins (reviewed in *Banani et al., 2017*; *Conti and Oppikofer, 2022*; *Sabari et al., 2020*; *Shin and Brangwynne, 2017*). The N-terminal region of UNG2 (amino acids 1–92) contains an IDR as calculated by Predictor of Natural Disordered Regions (PONDR) (*Figure 2C*) and previous reports indicate that the same region is partially required for UNG2's localization to laser-induced DNA damage (*Zeitlin et al., 2011*). Thus, we assessed whether UNG2's IDR allows UNG2 to form biomolecular condensates using the optoDroplet system, where an IDR of choice is fused to the photolyase homology region of *Arabidopsis thaliana* Cry2, a light-sensitive protein which self-associates upon blue light exposure (*Shin et al., 2017*). This method measures, in a tunable manner, the propensity of a given polypeptide to seed condensates in the absence of functional stimuli like DNA damage. We generated a panel of mutated UNG2 constructs fused to mCherry-Cry2, including full-length UNG2 (UNG2), UNG2 IDR only (IDR), UNG2 lacking the IDR (ΔIDR), UNG2 IDR moved to the C-terminus (IDR-C), and an empty vector (EV) containing only the mCherry-Cry2 construct (*Figure 2D*). UNG2 full-length, IDR, and IDR-C constructs expressed in UNG2 KO cells formed nuclear foci upon exposure to blue light, whereas the ΔIDR and EV constructs did not (*Figure 2E*). This suggests that UNG2's IDR is capable of seeding condensates. Within the IDR region of UNG2, there is a PCNA- and an RPA-binding domain. To assess the contribution of these domains, we also generated UNG2 constructs that lacked the PCNA interacting peptide (ΔPIP$^{F10,11A}$) or the ability to bind RPA (ΔRPA$^{R88C}$) and found that the ΔPIP construct formed nuclear foci upon exposure to blue light while the ΔRPA construct did not (*Figure 2E*).

To evaluate whether the propensity to form condensates was correlated to UNG2-dependent DNA repair, we explored the ability of the UNG2 constructs to complement the floxuridine sensitivity of UNG KO cells (*Figure 2—figure supplement 1H*). UNG KO cells expressing UNG2, IDR-C, ΔPIP, and

ΔRPA constructs rescued the sensitivity of UNG KO cells to floxuridine (*Figure 2—figure supplement 1H*), confirming that these constructs are functional. Interestingly, taken together with the optoDroplet data, this suggests that RPA binding may be required during S phase, but additional regions of the protein may compensate to direct repair during DNA damage conditions. The IDR-only construct lacks UNG's catalytic domain and was not able to complement floxuridine-induced cytotoxicity. UNG KO cells expressing the ΔIDR construct showed an intermediate phenotype (*Figure 2—figure supplement 1H*). These data suggest that UNG2's IDR can seed biomolecular condensates and is partially required for UNG2-dependent floxuridine repair.

Based on the premise that UNG2 can form biomolecular condensates, we used an in-cell proximity biotinylation assay to identify factors enriched in UNG2-IDR-seeded condensates (*De La Cruz et al., 2024*; *Lyons et al., 2023*). In this assay, a UNG2-IDR cDNA was fused to a LacI construct and biotinylating enzyme ascorbic acid peroxidase 2 (APEX2) and expressed in U2OS 2-6-3 cells. The LacI module allowed for the recruitment of multiple copies of UNG2-IDR to a particular genomic locus where a LacO array is integrated, resulting in high local concentrations of UNG2-IDR that are sufficient to seed a condensate (*Figure 2F*). At this locus, factors present in UNG2-IDR-seeded condensates were biotinylated by APEX2, purified, and identified by LC–MS/MS (*Figure 2F*; *Hung et al., 2016*; *Lam et al., 2015*). As expected, we identified peptides mapping to UNG2-IDR itself (*Figure 2G*, *Supplementary file 3*). Proteins that were present in UNG2-IDR-seeded condensates were also enriched in DNA repair and replication pathways (*Figure 2—figure supplement 1I*). This included downstream repair factors in BER, such as LIG1, and MMR factors such as MSH2, as well as single-stranded binding protein, and RPA2, a known UNG interacting partner (*Hagen et al., 2008*; *Kavli et al., 2021*; *Torseth et al., 2012*; *Figure 2G*, *Supplementary file 3*). The recruitment of DNA repair factors by UNG2-IDR-seeded condensates occurred in the absence of UNG2's catalytic domain and uracil-based damage, suggesting that UNG2-IDR is sufficient to recruit repair factors relevant for U-BER. Importantly, peptides for both METTL3 and METTL14 were also identified as enriched in UNG2-IDR-seeded condensates (*Figure 2G*, *Supplementary file 3*). This provides additional evidence that N6-adenosine methyltransferases may cooperate with UNG2 in uracil repair.

## METTL3 deposits 6mA in DNA in response to agents that increase genomic uracil

Given the identification of N6-adenosine methyltransferases in orthogonal discovery-based methods related to repair of uracil-containing DNA, we examined the contribution of METTL3 and its substrate, N6-methyladenosine, to uracil repair. Consistent with the screen results obtained in MMR-proficient HT-29 cells (*Figure 1A, B*), METTL3 KO sensitized DLD-1 cells, an MMR-deficient CRC cell line, to floxuridine (*Figure 3A, B*, *Figure 3—source data 1*). To determine whether the methyltransferase activity of METTL3 is important for its role in uracil repair, we assessed the sensitivity of DLD-1 and SW620 cells to floxuridine in the absence or presence of a tool inhibitor for METTL3 (*Yankova et al., 2021*). Treatment with the METTL3 inhibitor increased floxuridine sensitivity in both cell lines, suggesting that METTL3 methyltransferase activity is important for its function in repair of floxuridine-induced lesions (*Figure 3C*, *Figure 3—figure supplement 1A*). We also found that knockout of WTAP sensitized cells to floxuridine, but to a lesser extent than knockout of METTL3 (*Figure 3—figure supplement 1B, C*, *Figure 3—figure supplement 1—source data 1*).

We next observed the deposition of N6-methyladenosine in response to an increase in genomic uracil. The methylation modification can be visualized using an antibody that recognizes N6-methyladenosine in single-stranded nucleic acid species, whether RNA or DNA. Pre-extraction of cells prior to fixation allows for the removal of cytoplasmic content, reducing background signal from m6A-modified RNA in the cytoplasm (*Figure 3—figure supplement 1D*). Using non-denaturing conditions, we observed minimal N6-methyladenosine signal in untreated cells. Upon treatment with floxuridine, we observed a significant increase of cells displaying nuclear N6-methyladenosine foci (*Figure 3D, E*). DNase and RNase treatments prior to staining allowed for discrimination of DNA or RNA as the nucleic acid species that produced the N6-methyladenosine foci. The percentage of cells that displayed floxuridine-induced N6-methyladenosine foci were unchanged in response to treatment with RNase A (*Figure 3D, E*). Using an agarose gel, we confirmed that the RNase A treatment efficiently removed RNA (*Figure 3—figure supplement 1E*). Treatment with RNase H, which degrades the RNA strand of RNA–DNA duplexes, also failed to reduce the percentage of cells with

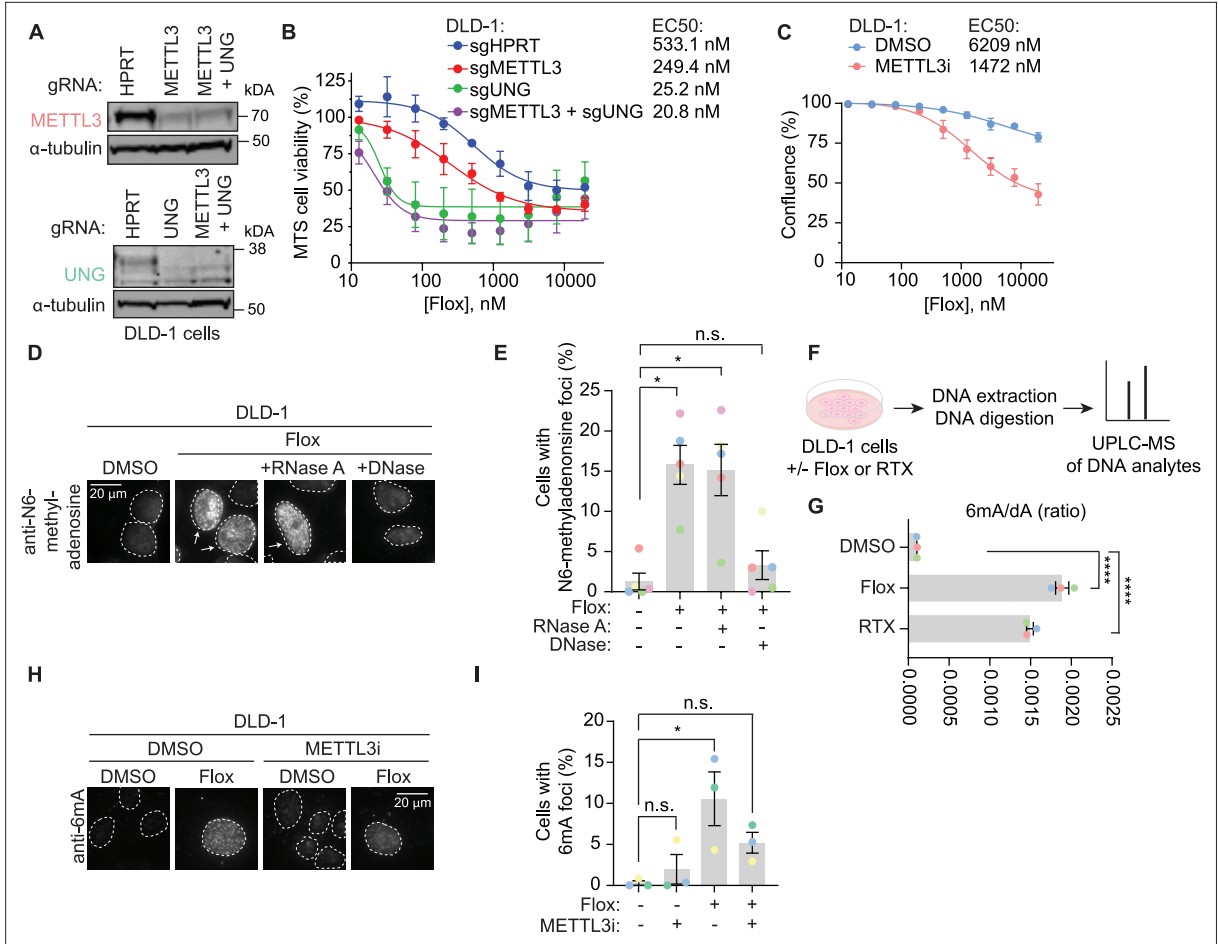

**Figure 3.** METTL3 deposits N6-methyladenosine in DNA in response to agents that increase genomic uracil. (**A**) Representative immunoblot images from DLD-1 cells nucleofected with ribonucleoproteins containing Cas9 and indicated guide RNAs (gRNA) as performed for B. α-Tubulin represents loading control. (**B**) MTS cell viability assay in the presence of floxuridine. Error bars, mean ± SEM, *n* = 2 biological replicates. (**C**) Growth curves in DLD-1 cells upon treatment with 15 μM METTL3 inhibitor and indicated drug concentration post-treatment. Error bars, mean ± SEM, *n* = 3 biological replicates. (**D**) Representative images from DLD-1 UNG KO upon treatment with floxuridine for 66 hr. Prior to staining with N6-methyladenosine antibody, indicated samples were treated with RNase A or DNase. (**E**) Quantification of experiment represented in D for a percentage of cells with >5 N6-methyladenosine foci. Error bars, mean ± SEM; repeated measures one-way ANOVA with Dunnet's multiple comparisons test with a single pooled variance, *p ≤ 0.05, *n* = 5 biological replicates. (**F**) Schematic of the experiment shown in G. DLD-1 cells were treated with DMSO, 500 nM floxuridine, or 500 nM raltitrexed for 72 hr. Cells were washed, collected, and DNA was purified and digested with DNA degradase plus enzyme prior to separation and quantification by ultra-performance liquid chromatography–mass spectrometry (UPLC–MS/MS). (**G**) The ratio of 6mA analyte to dA analyte as detected in DNA of DLD-1 cells upon treatment with 500 nM floxuridine or 500 nM raltitrexed using UPLC–MS/MS. ****p ≤ 0.0001, *n* = 3 biological replicates. (**H**) Representative images of 6mA staining in DLD-1 UNG KO cells upon treatment with 500 nM of floxuridine and 30 μM METTL3 inhibitor at 64 hr. (**I**) Quantification of experiment represented in G for a percentage of cells with >10 6mA foci. Error bars, mean ± SEM; ordinary one-way ANOVA with Dunnett's multiple comparisons test with a single pooled variance, *p ≤ 0.05, *n* = 3 biological replicates. HPRT, cutting control targeting intronic region of HPRT gene; Flox, floxuridine; RTX, raltitrexed; METTL3i, METTL3 inhibitor.

The online version of this article includes the following source data and figure supplement(s) for figure 3:

**Source data 1.** Source data for panels B, E and I.

**Source data 2.** Source data for panel A.

**Figure supplement 1.** N6-methyladenosine foci in response to genomic uracil-inducing agents are not RNA:DNA hybrids.

**Figure supplement 1—source data 1.** Source data for panels A, G, H.

**Figure supplement 1—source data 2.** Source data for panel B.

floxuridine-induced N6-methyladenosine foci (*Figure 3—figure supplement 1F, G*). In contrast, upon treatment with DNase, the percentage of cells with floxuridine-induced N6-methyladenosine foci was significantly reduced and became statistically equivalent to untreated conditions, indicating that the observed N6-methyladenosine species is a modification of DNA, 6mA. Removal of the DAPI signal upon DNase treatment confirmed effective DNA degradation (*Figure 3—figure supplement 1H*). These results were confirmed with an orthogonal approach using mass spectrometry to identify 6mA and dA analytes in purified DNA (*Figure 3F*). This approach does not rely on an antibody to detect 6mA and can conclusively distinguish between DNA and RNA analytes based on exact mass. Using ultra-performance liquid chromatography coupled to tandem mass spectrometry (UPLC–MS/MS), we observed that the ratio of 6mA to dA significantly increased with floxuridine treatment (*Figure 3G*). Similar results were observed in cells treated with raltitrexed (*Figure 3G*, *Figure 3—figure supplement 1I, J*).

METTL3, together with METTL14, can methylate DNA in purified settings (*Qi et al., 2022*; *Woodcock et al., 2019*; *Yu et al., 2021*) and contributes to the trace amounts of 6mA observed at baseline in gDNA (*Chen et al., 2022a*). We hypothesized that 6mA foci observed in response to floxuridine treatment were deposited by METTL3. Consistent with this hypothesis, when cells were co-treated with a METTL3 inhibitor, floxuridine or raltitrexed treatment did not significantly change the percentage of 6mA foci compared to the DMSO control (*Figure 3H, I*; *Figure 3—figure supplement 1K, L*), demonstrating that 6mA foci linked to uracil incorporation in DNA are at least partially METTL3 dependent.

## 6mA promotes uracil repair upstream of UNG2 in U-BER

Our data show that METTL3-dependent deposition of 6mA occurs in response to uracil-based DNA damage, which is predominantly repaired by UNG-dependent mechanisms. Thus, we sought to determine the molecular relationship between METTL3 and 6mA with UNG2. First, we tested whether METTL3 inhibition could modulate UNG2 foci formation following uracil-based DNA damage. We observed that METTL3 inhibition alone causes an increase in UNG2 foci in untreated conditions (no floxuridine) (*Figure 4A, B*). This is likely explained by increased expression of UNG2-mCherry as observed by qPCR and western blot (*Figure 4C*, *Figure 4—figure supplement 1*, *Figure 4—figure supplement 1—source data 1*), which is consistent with the described dependence of condensate formation on protein concentration (*Conti and Oppikofer, 2022*; *Sabari et al., 2020*). Since METTL3 can methylate mRNAs to affect their stability, it is possible that METTL3 is altering the stability of the mCherry-UNG2 transcripts in an RNA-dependent mechanism. As described in *Figure 1A*, treatment with floxuridine alone increased the percentage of cells with UNG2 foci further than that of METTL3 inhibition alone (*Figure 4A, B*). Importantly, concomitant treatment with floxuridine and a METTL3 inhibitor significantly reduced the percentage of cells displaying UNG2 foci compared to the floxuridine only conditions (*Figure 4A, B*). Inversely, we tested whether UNG2 was required to form 6mA foci in response to floxuridine treatment. The percentage of cells that formed 6mA foci in response to floxuridine treatment was unchanged in wild-type and UNG KO DLD-1 cells (*Figure 4D, E*). These data are consistent with METTL3 and 6mA functioning upstream of UNG2 in repair of uracil-based DNA lesions.

To understand if 6mA embedded in DNA has a direct effect on modulating UNG loading onto DNA, we examined the binding kinetics of recombinant UNG catalytic domain (amino acids 93–313, present in both UNG isoforms) to 6mA-containing DNA templates by biolayer interferometry. We did not observe changes in UNG binding between dsDNA containing 6mA, either when 6mA was base paired with uracil or when 6mA was present two bases away from uracil on the same strand, and ssDNA containing 6mA compared to uracil only containing controls (*Figure 4—figure supplement 1B, C*).

## 6mA promotes genome repair of base damage beyond uracil incorporation

Building on the importance of METTL3-dependent 6mA modification in response to floxuridine treatment, we sought to define whether 6mA's role in DNA repair is unique to uracil or a general response to DNA damage. To do so, we examined the ability of METTL3 inhibition to sensitize SW620 cells to a panel of DNA-damaging agents. We found METTL3 inhibition sensitized cells to HU and gemcitabine, but minimally to MMC (*Figure 5A–C*). Similarly, the percentage of cells with 6mA foci was significantly

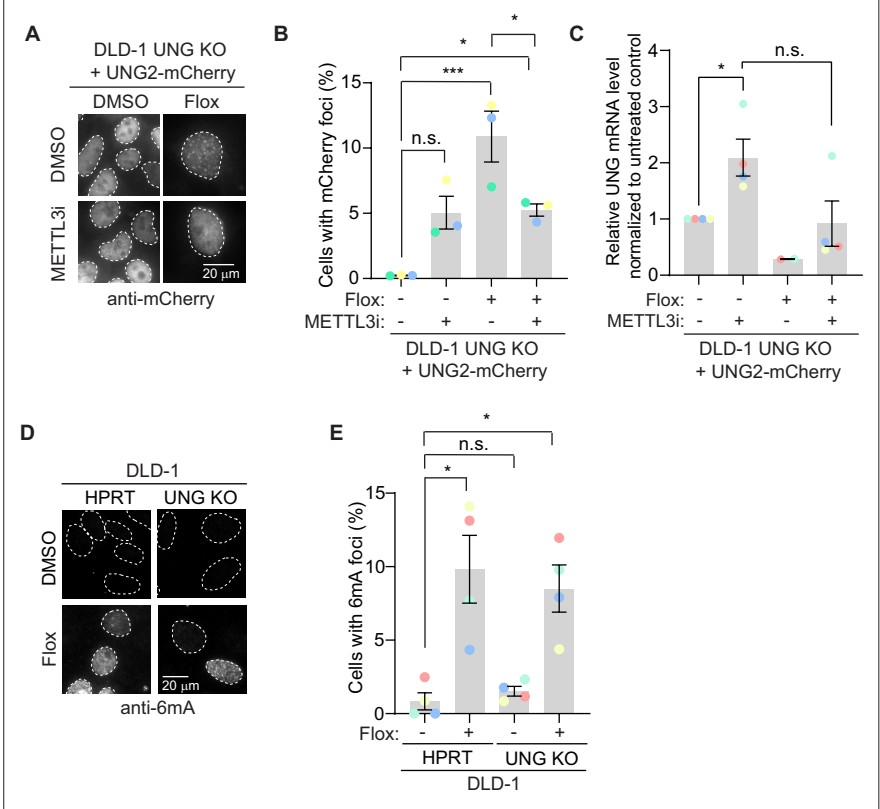

**Figure 4.** 6mA promotes uracil repair upstream of UNG2 in uracil base excision repair. (**A**) Representative images of mCherry staining in DLD-1 UNG KO cells expressing UNG2-mCherry cDNAs upon treatment with 500 nM floxuridine and 30 µM METTL3 inhibitor at 64 hr. (**B**) Quantification of experiment represented in A for percentage of cells with >5 mCherry foci. Error bars, mean ± SEM; ordinary one-way ANOVA with Tukey's multiple comparisons test with a single pooled variance, *p ≤ 0.05, ***p ≤ 0.001, n = 3 biological replicates. (**C**) Real-time quantitative PCR from A, B for UNG2 transcript levels normalized to tubulin controls. Error bars, mean ± SEM; Mann–Whitney *t*-test for the following pairs: DMSO versus METTL3 inhibitor and floxuridine versus floxuridine + METTL3 inhibitor. *p ≤ 0.05, n = 3 biological replicates except for the floxuridine only condition which includes n = 2 biological replicates. (**D**) Representative images of 6mA staining in DLD-1 UNG KO cells upon treatment with 500 nM floxuridine at 64 hr. HPRT indicates wild-type cells. These cells were targeted with a cutting control targeting the intronic region of HPRT gene. (**E**) Quantification of experiment represented in D for percentage of cells with >5 6 mA foci. Error bars, mean ± SEM; RM one-way ANOVA with Dunnet's multiple comparisons test with a single pooled variance, *p ≤ 0.05, n = 5 biological replicates. Flox, floxuridine; METTL3i, METTL3 inhibitor.

The online version of this article includes the following source data and figure supplement(s) for figure 4:

**Source data 1.** Source data for panels B, C, and E.

**Figure supplement 1.** The presence of 6mA does not alter UNG-binding kinetics.

**Figure supplement 1—source data 1.** Source data for panel A.

increased in response to HU and gemcitabine, but not MMC (*Figure 5D, E*). To assess whether DNA damage levels were linked to the ability of cells to form 6mA foci, we quantified the mean nuclear γH2AX intensity for each treatment. MMC treatment induced similar γH2AX levels to those produced by treatment with floxuridine or raltitrexed (*Figure 5—figure supplement 1A, B*). Thus, the occurrence of 6mA foci does not seem to be a general response to damage, such as the MMC-induced replication stress, but an event linked to base DNA damage. This base damage extends beyond that of only uracil DNA incorporation since gemcitabine does not cause uracil incorporation into the genome and accordingly does not induce UNG2 foci (*Figure 2—figure supplement 1C, D*). Further, when cells are co-stained for both mCherry and 6mA, there are cases where UNG2 and 6mA form foci in the same cells and appear to have overlapping signals (*Figure 5—figure supplement 1C*).

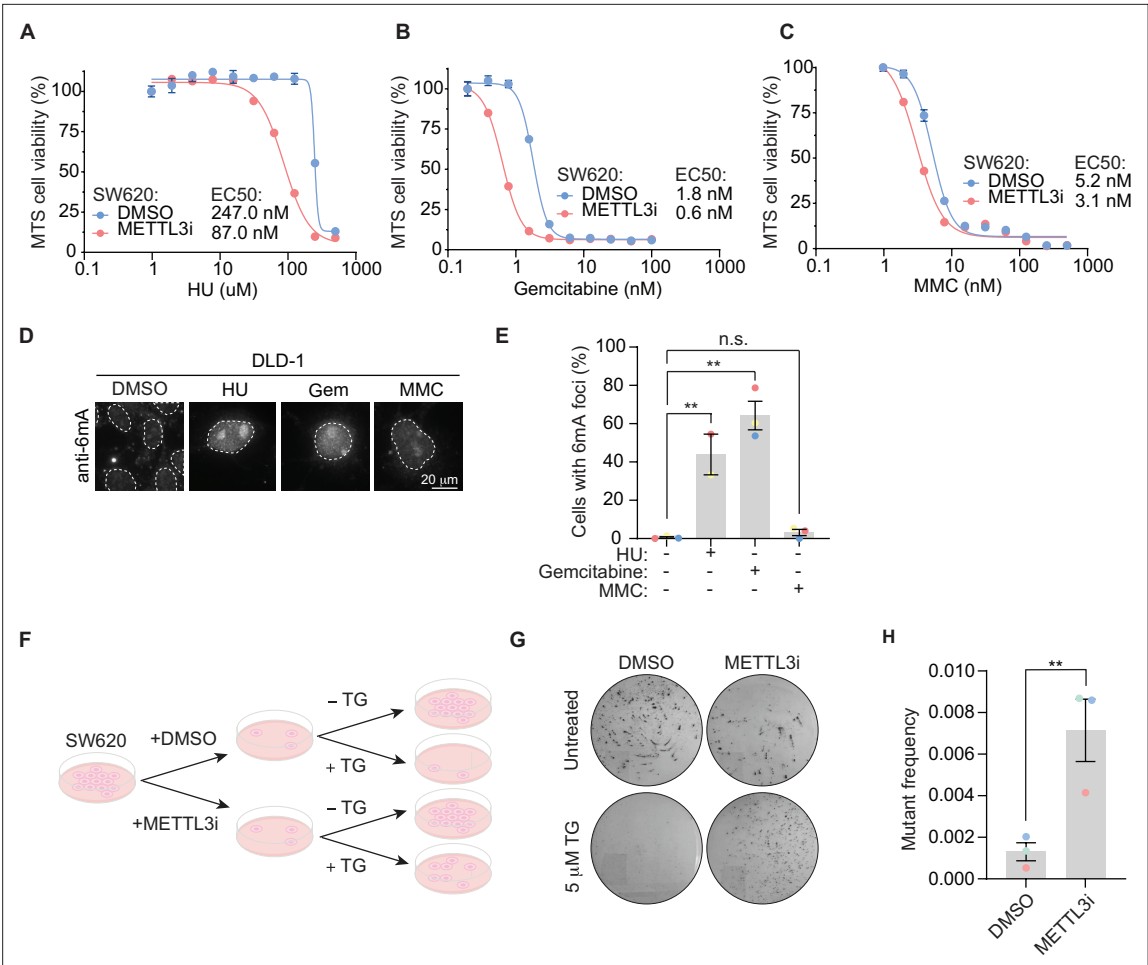

**Figure 5.** 6mA promotes genome repair of base damage beyond uracil incorporation. (**A–C**) MTS cell viability in SW620 cells upon treatment with METTL3 inhibitor and indicated concentrations of drugs. Error bars, mean ± SEM, $n = 3$, technical replicates, representative of three biological replicates. (**D**) Representative images of 6mA staining in DLD-1 UNG KO cells upon treatment with indicated DNA damaging agents at 64 hr. (**E**) Quantification of experiment represented in D for percentage of cells with >5 6 mA foci. Error bars, mean ± SEM; ordinary one-way ANOVA with Dunnet's multiple comparisons test with a single pooled variance, **$p \leq 0.01$, $n = 3$ biological replicates for all except for the HU condition which includes $n = 2$ biological replicates. (**F**) Schematic of colony formation assay. SW620 cells, maintained in HAT media, were treated with 30 μM of METTL3 inhibitor for 7 days. $0.5 \times 10^4$ METTL3 inhibitor-treated cells seeded in the presence of 5 μM 6-thioguanine (TG) and colony formation assay was assessed after 14 days. $0.5 \times 10^2$ METTL3 inhibitor-treated cells were seeded for untreated controls. (**G**) Representative dishes after 14 days of growth in 5 μM TG as described in F. (**H**) Quantitation of mutation frequency from G. Mutation frequency was calculated by normalizing to the untreated controls. Error bars, mean ± SEM; paired *t*-test, **$p \leq 0.01$, $n = 3$ biological replicates. HU, hydroxyurea; Gem, gemcitabine; MMC, mitomycin C; METTL3i, METTL3 inhibitor.

The online version of this article includes the following source data and figure supplement(s) for figure 5:

**Source data 1.** Source data for panels A–C and E.

**Figure supplement 1.** 6mA foci, but not UNG2 foci, correlate with DNA damage levels.

**Figure supplement 1—source data 1.** Source data for panel B.

However, there are cases where 6mA forms foci and UNG2 does not (**Figure 5—figure supplement 1C**).

Our data suggest that 6mA embedded in DNA facilitates repair of lesions caused by DNA-damaging agents beyond that of uracil. Importantly, 6mA has an established role in MMR in prokaryotes that has not been described for eukaryotes. In *Escherichia coli*, dam methylase labels the parental strand of plasmid DNA with 6mA during replication. This results in hemi-methylated DNA indicating to the DNA repair machinery the parental strand (bearing 6mA modifications) that acts as the template for repair and the daughter strand that needs repairing. 6mA provides the strand discrimination signal

for repair of mismatched bases on the daughter strand in *E. coli*. To explore whether there exists an evolutionarily conserved function for 6mA in MMR, we examined the relationship between METTL3 and MMR in mammalian cells. We performed an MMR assay measuring mutation frequency in SW620 cells, a human MMR-proficient cell line. A deficiency in MMR results in increased mutation frequency, including at the HPRT locus leading to resistance to 6-thioguanine (TG), as previously described (*Glaab et al., 1998*; *McCormick and Maher, 1981*). Upon pre-treatment with a METTL3 inhibitor, SW620 cells were more resistant to TG, reflecting an increase in damaging mutations at the HPRT locus and overall mutational burden (*Figure 5I–K*). These data suggest that METTL3 promotes MMR-dependent DNA repair in mammalian cells.

## Discussion

This study demonstrates that 6mA deposition in DNA is functionally relevant to DNA damage repair in mammalian cells with two orthogonal methods. The first, single-cell imaging, uses an antibody and various nucleases to demonstrate the N6-methyladenosine foci are DNA. Treatment with DNase removes the foci while RNase A and RNase H do not. The second approach, UPLC–MS/MS of purified DNA, does not rely on the use of an antibody and can discriminate DNA signals by mass. Prior investigations on 6mA deposition in DNA have been controversial. A potential cause of artifact 6mA signals in mammalian cell lines is bacterial contamination (reviewed in *Feng and He, 2023*). The baseline levels of 6mA in cell lines are low and any contamination with bacteria can produce an erroneous signal. In our study, cells only show the 6mA signal when treated with DNA-damaging agents, and the 6mA is absent from untreated cells (*Figure 3D–F*), suggesting that the 6mA signal is not a result of bacterial contamination. Moreover, our cell lines routinely tested negative for mycoplasma contamination. It could be possible that stock solutions of DNA damaging agents may be contaminated, but this would need to be true for all individual drugs and stocks tested. Moreover, the data showing 6mA signal is not significantly different from untreated cells when a DNA damaging agent is combined with a METTL3 inhibitor (*Figure 3G, H*) provides strong evidence against bacterial contamination in our stocks. Additionally, only 10–20% of treated cells show 6mA foci, which also argues against widespread contamination.

Our data provide evidence that 6mA deposition during DNA repair is METTL3 dependent. While 'writers' of 6mA in mammalian cells are not well defined, there is literature showing that METTL3 can methylate DNA. METTL3 methylates ssDNA or mismatched dsDNA in biochemical experiments (*Yu et al., 2021*). Intriguingly, METTL3 methylates DNA with higher efficiency compared to RNA, while the reverse pattern is observed with regard to binding affinity, with METTL3 binding with higher affinity to RNA (*Qi et al., 2022*). In cells, METTL3 contributes to low baseline 6mA levels in the genome (*Chen et al., 2022a*). Our data, for the first time, endow 6mA with a function, namely the repair of base lesions, and the identification of 6mA in the repair of DNA base damage has potential therapeutic implications. It is possible that combining METTL3 inhibition with a base-damaging chemotherapeutic agent would improve the efficacy of chemotherapy alone. Additionally, loss of METTL3 has been shown to enhance response to anti-PD-1 treatment in MMR proficient, low microsatellite instability CRC (*Wang et al., 2020*) as well as in lung tumor models (*Yu et al., 2024*). Given the relationship between MMR status, microsatellite instability, and cancer immunotherapies, we hypothesize that loss of factors that deposit 6mA may also serve as a potential biomarker for dysregulated MMR. Thus, the discovery presented here of a functional role for 6mA in promoting genome stability represents a significant advancement from both fundamental and clinical perspectives.

### Limitations of the study

We did not see an effect of 6mA embedded in DNA on UNG binding in purified settings. However, the biochemical assay used presents certain limitations. For instance, there could be additional factors present in the cell, but excluded from our purified system, that impact the ability of 6mA to modulate UNG binding. Proteins that bind m6A in RNA, often referred to as 'readers', have various roles that direct the biological function of the modification. While this hypothesis remains to be assessed, EIF3A and HNRNPC, known m6A RNA readers, scored in the UNG2-IDR condensate proximity biotinylation assay described in *Figure 2*. It is possible that UNG recognizes specific placements of 6mA in relation to U, or structural features, such as a forked substrate representing DNA replication, not tested in

our binding assay. Our assay would also not be able to detect if other UNG domains, outside of the catalytic domain, alter binding to the 6mA-containing DNA templates.

While we uncovered a role for METTL3 in depositing 6mA in the genome, we did not reveal the mechanism that controls 6mA incorporation in gDNA. The increased presence of 6mA during DNA damage could result from methylation at a pre-existing unmodified A base already present within DNA or from incorporation of pre-modified 6mA during DNA synthesis, as observed in experiments observing Pol $\lambda$-dependent genomic incorporation of supplemented N6-methyldeoxyadenosine in mammalian cells (*Liu et al., 2021*). Our data do not discriminate between these two mechanisms.

Lastly, it is possible other methyltransferase, such as METTL4 and N6AMT1, both of which have been shown to introduce 6mA in DNA, could contribute to 6mA deposition in response to DNA damage or other stimuli (*Chen et al., 2022b*; *Hsu et al., 2022*; *Kweon et al., 2019*; *Sheng et al., 2020*; *Xiao et al., 2018*). While we established the importance of the N6-adenosine methyltransferase, METTL3, we did not explore a potential role for 6mA demethylases, such as ALKBH1 (*Wang et al., 2023*) or ALKBH4 (*Kweon et al., 2019*; *Zhang et al., 2020*). We note that FTO, an RNA m6A demethylase, scored in the UNG2-IDR condensate proximity biotinylation assay (*Figure 2*). We imagine there would exist a finely tuned balance between methylation and demethylation in regulating 6mA levels and function.

# Materials and methods

**Key resources table**

| Reagent type (species) or resource | Designation | Source or reference | Identifiers | Additional information |
|---|---|---|---|---|
| Cell line (*Homo sapiens*) | Human: DLD-1 | ATCC | Catalog # CCL-221, RRID:CVCL_0248 | |
| Cell line (*Homo sapiens*) | Human: HT-29 | ATCC | Catalog # HTB-38, RRID:CVCL_0320 | |
| Cell line (*Homo sapiens*) | Human: SW620 | ATCC | Catalog # CCL-227, RRID:CVCL_0547 | |
| Cell line (*Homo sapiens*) | Human: U2OS 2-6-3 | Spector Lab | PMID:15006351 | |
| Transfected construct (*Homo sapiens*) | Human: DLD-1 UNG KO Clone E7 | This paper | This paper | Pfizer, Inc |
| Transfected construct (*Homo sapiens*) | Human: DLD-1 UNG KO Clone E7 + pMCS_UNG2_AID_mCherry | This paper | This paper | Pfizer, Inc |
| Transfected construct (*Homo sapiens*) | Human: DLD-1 UNG KO Clone E7 + pMCS_AID_mCherry | This paper | This paper | Pfizer, Inc |
| Transfected construct (*Homo sapiens*) | Human: DLD-1 UNG KO Clone E7 + pCMV_Cry2_mCherry_EV | This paper | This paper | Pfizer, Inc |
| Transfected construct (*Homo sapiens*) | Human: DLD-1 UNG KO Clone E7 + pCMV_Cry2_mCherry_UNG2 | This paper | This paper | Pfizer, Inc |
| Transfected construct (*Homo sapiens*) | Human: DLD-1 UNG KO Clone E7 + pCMV_Cry2_mCherry_UNG2_IDR | This paper | This paper | Pfizer, Inc |
| Transfected construct (*Homo sapiens*) | Human: DLD-1 UNG KO Clone E7 + pCMV_Cry2_mCherry_UNG2_DIDR | This paper | This paper | Pfizer, Inc |
| Transfected construct (*Homo sapiens*) | Human: DLD-1 UNG KO Clone E7 + pCMV_Cry2_mCherry_UNG2_IDR-C | This paper | This paper | Pfizer, Inc |

*Continued on next page*

*Continued*

| Reagent type (species) or resource | Designation | Source or reference | Identifiers | Additional information |
|---|---|---|---|---|
| Transfected construct (*Homo sapiens*) | Human: DLD-1 UNG KO Clone E7 + pCMV_Cry2_ mCherry_UNG2_ DPIP | This paper | This paper | Pfizer, Inc |
| Transfected construct (*Homo sapiens*) | Human: DLD-1 UNG KO Clone E7 + pCMV_Cry2_ mCherry_UNG2_ DRPA | This paper | This paper | Pfizer, Inc |
| Transfected construct (*Homo sapiens*) | Human: HT-29 pLenti7-EF1a-Cas9 | This paper | This paper | Pfizer, Inc |
| Transfected construct (*Homo sapiens*) | Human: U2OS 2-6–3+GFP-LacI-APEX2-UNG2$^{IDR}$ | This paper | This paper | Pfizer, Inc |
| Recombinant DNA reagent | pCMV-Gag-Pol | CellBioLabs | Catalog # RV-111 | Sabari Lab |
| Recombinant DNA reagent | pCMV-VSV-G | CellBioLabs | Catalog # RV-110 | |
| Recombinant DNA reagent | pLenti-EF1a-Cas9 | Pfizer | #5342 | |
| Recombinant DNA reagent | Custom sgRNA library | DeskGen | This paper | |
| Recombinant DNA reagent | pMCS-Puro Retroviral Vector | CellBio Labs | Catalog # RTV-041 | Pfizer, Inc |
| Recombinant DNA reagent | pMCS_AID_mCherry_Puro Retroviral Vector | Azenta Life Sciences | N/A – C096 – see ***Supplementary file 4*** | |
| Recombinant DNA reagent | pMCS_UNG2_AID_mCherry Retroviral Vector | Azenta Life Sciences | N/A – C096 – see ***Supplementary file 4*** | |
| Recombinant DNA reagent | pLenti-GIII-CMV | Applied Biological Materials | 16422061 | |
| Recombinant DNA reagent | pLenti-CMV-Cry2-mCherry-SV40-Puro (EV) Lentiviral Vector | Applied Biological Materials | N/A – C096 – see ***Supplementary file 4*** | |
| Recombinant DNA reagent | pLenti-CMV-Cry2-mCherry-UNG2-SV40-Puro Lentiviral Vector | Applied Biological Materials | N/A – C096 – see ***Supplementary file 4*** | |
| Recombinant DNA reagent | pLenti-CMV-Cry2-mCherry-UNG2-IDR-SV40-Puro Lentiviral Vector | Applied Biological Materials | N/A – C096 – see ***Supplementary file 4*** | |
| Recombinant DNA reagent | pLenti-CMV-Cry2-mCherry-UNG2-DIDR-SV40-Puro Lentiviral Vector | Applied Biological Materials | N/A – C096 – see ***Supplementary file 4*** | |
| Recombinant DNA reagent | pLenti-CMV-Cry2-mCherry-UNG2-IDR-C-SV40-Puro Lentiviral Vector | Applied Biological Materials | N/A – C096 – see ***Supplementary file 4*** | |
| Recombinant DNA reagent | pLenti-CMV-Cry2-mCherry-UNG2-DPIP-SV40-Puro Lentiviral Vector | Applied Biological Materials | N/A – C096 – see ***Supplementary file 4*** | |
| Recombinant DNA reagent | pLenti-CMV-Cry2-mCherry- DRPA-SV40-Puro Lentiviral Vector | Applied Biological Materials | N/A – C096 – see ***Supplementary file 4*** | |
| Antibody | Anti-N6-methyladenosine (anti-6mA), rabbit polyclonal | Synaptic System | Catalog # 202 003, RRID:AB_2279214 | 1:100 IF |
| Antibody | Anti-phospho-Histone H2A.X (Ser139). Clone JBW301, mouse monoclonal | Millipore | 05-636; RRID:AB_309864 | 1:1000 IF |
| Antibody | Anti-mCherry, chicken polyclonal | Abcam | Catalog # Ab205402, RRID:AB_2722769 | 1:500 IF |
| Antibody | Anti-mCherry, recombinant rabbit | Abcam | Catalog # Ab213511; RRID:AB_2814891 | 1:500 IF |
| Antibody | Alexa Fluor 488-conjugated Anti-mouse IgG (H+L), goat polyclonal | Invitrogen | Catalog # A11029, RRID:AB_2534088 | 1:1000 IF |

*Continued on next page*

*Continued*

| Reagent type (species) or resource | Designation | Source or reference | Identifiers | Additional information |
|---|---|---|---|---|
| Antibody | Alexan Fluor 488-conjugated anti-Chicken IgY (H+L), goat polyclonal | Invitrogen | Catalog # A32931, AB_2762843 | 1:1000 IF |
| Antibody | Anti-UNG, rabbit polyclonal | AbClonal | Catalog # A1261 (WB: 1:1000); RRID:AB_2759453 | 1:500 WB |
| Antibody | Anti-METTL3, rabbit polyclonal | AbClonal | Catalog # 8370 (WB: 1:1000); RRID:AB_2770344 | 1:500 WB |
| Antibody | Anti-Tubulin, Clone DM1A (mouse), rabbit monoclonal | Millipore | Catalog # MABT205; RRID:AB_11204167 | 1:1000 WB |
| Antibody | Anti-WTAP, rabbit polyclonal | Bethyl | Catalog # A301-435A; RRID:AB_961137 | 1:500 WB |
| Antibody | IRDye 680RD Goat anti-rabbit, goat polyclonal | Licor | Catalog # 926-68071; RRID:AB_10956166 | 1:10,000 WB |
| Antibody | IRDye 800CW Donkey anti-mouse, donkey unknown clonality | Licor | Catalog # 926-32212, RRID:AB_621847 | 1:10,000 WB |
| Antibody | IgG, rabbit monoclonal | Abcam | Catalog # Ab172730, RRID_2687931 | IP: 5 µg |
| Antibody | UNG, rabbit polyclonal | Abclonal | Catalog # A1261; RRID:AB_2759453 | 1:500 WB, IP: 5 µg |
| Sequence-based reagent | gRNA_UNG_4, CTTGATGGGCACGAACCGTG | IDT | N/A | |
| Sequence-based reagent | gRNA_HPRT, AATTATGGGGATTACTAGGA | IDT | N/A; targets intronic region | |
| Sequence-based reagent | gRNA_METTL3-ex10-1, CAGTTGGGGTTGCACATTGTG | IDT | N/A | |
| Sequence-based reagent | gRNA_UNG-2597, TCCCCTTTGTCAGTGTATAG | IDT | N/A | |
| Sequence-based reagent | gRNA_WTAP_Hs.Cas9.WTAP.1.AB | IDT | Catalog # 313817305 | |
| Sequence-based reagent | gRNA_METTL3_ Hs.Cas9.METTL3.1.AA | IDT | Catalog # 313817302 | |
| Sequence-based reagent | gRNA_NTC GTAGCGAACGTGTCCGGCGT | IDT | N/A | |
| Sequence-based reagent | dsDNA-U:A, 5'-/5Biosg//iSp9/AAATTGUTATCCGCT Complement: 5'-AGCGGATAACAATTT | IDT | N/A | |
| Sequence-based reagent | dsDNA-U:m6dA, 5'-/5Biosg//iSp9/ AAATTGUTATCCGCT Complement: 5'- AGCGGATA/iN6Me-dA/CAATTT | IDT | N/A | |
| Sequence-based reagent | dsDNA-U:A, m6dA:T, 5'-/5Biosg//iSp9/AAATTGUT/iN6Me-dA/TCCGCT Complement: 5'-AGCGGATA ACAATTT | IDT | N/A | |
| Sequence-based reagent | ssDNA-U: /5Biosg//iSp9/AAATTGUTATCCGCT | IDT | N/A | |
| Sequence-based reagent | ssDNA-U_m6dA: /5Biosg//iSp9/AAATTGUT/iN6Me-dA/TCCGCT | IDT | N/A | |
| Peptide, recombinant protein | Recombinant UNG-Catalytic Domain | This paper | This paper | Pfizer, Inc |
| Commercial assay or kit | QIAquick PCR Cleanup Kit | QIAGEN | Catalog # 28506 | |
| Commercial assay or kit | PureLink Quick PCR Purification Kit | Invitrogen | Catalog # K310001 | |
| Commercial assay or kit | Gentra Puregene kit | QIAGEN | Catalog # 158845 | |

*Continued on next page*

*Continued*

| Reagent type (species) or resource | Designation | Source or reference | Identifiers | Additional information |
|---|---|---|---|---|
| Commercial assay or kit | MTS Assay kit | Abcam | Catalog # ab197010 | |
| Commercial assay or kit | Quick-DNA/RNA Miniprep Plus Kit | Zymo Research | Catalog # D7003 | |
| Commercial assay or kit | RNeasy Plus University Kit | QIAGEN | Catalog # 730404 | |
| Commercial assay or kit | High Capacity RT Kit | Applied Biosystems | Catalog # 4374966 | |
| Chemical compound, drug | Phosphate Buffered Saline (PBS) | Corning | Catalog # 21-040-CV | |
| Chemical compound, drug | Heat Inactivated Fetal Bovine Serum (FBS) | Gibco | Catalog # 16140-071 | |
| Chemical compound, drug | RPMI-1640 | Corning | Catalog # 10-040-CM | |
| Chemical compound, drug | McCoy's 5A | Gibco | Catalog # 16600-108 | |
| Chemical compound, drug | DMEM | Gibco | Catalog # 11995073 | |
| Chemical compound, drug | DMEM (No Phenol Red) | Gibco | Catalog # A1443001 | |
| Chemical compound, drug | Penicillin Streptomycin Solution, 100× | Corning | Catalog # 30-002-CI | |
| Chemical compound, drug | Penicillin Streptomycin Solution | Gibco | Catalog # 15120-122; used for U2OS 2-6-3 | |
| Chemical compound, drug | GlutaMax | Gibco | Catalog # 35050061 | |
| Chemical compound, drug | 0.25% Trypsin | Corning | Catalog # 25-053-CI | |
| Chemical compound, drug | Recovery Cell Culture Freezing Medium | Gibco | Catalog # 12648010 | |
| Chemical compound, drug | HAT Supplement | Gibco | Catalog # 21060-017 | |
| Chemical compound, drug | Floxuridine | Sigma-Aldrich | Catalog # F0503 | |
| Chemical compound, drug | Raltitrexed | Sigma-Aldrich | Catalog # R9156 | |
| Chemical compound, drug | Gemcitabine | Sigma-Aldrich | Catalog # G6423 | |
| Chemical compound, drug | Hydroxyurea | Usp | Catalog # 1332000 | |
| Chemical compound, drug | Mitomycin C | StemCell Technologies | Catalog # 73273 | |
| Chemical compound, drug | METTL3 inhibitor | MedChem Express | Catalog # HY-134836/CS-0159584 | |
| Chemical compound, drug | Puromycin | Thermo Scientific | Catalog # J67236.XF | |
| Chemical compound, drug | Hygromycin B | Thermo Fisher | Catalog # 10687010 | |
| Chemical compound, drug | 6-Thioguanine | Tocris | Catalog # 4061 | |

*Continued on next page*

*Continued*

| Reagent type (species) or resource | Designation | Source or reference | Identifiers | Additional information |
|---|---|---|---|---|
| Chemical compound, drug | Lentiviral Packaging Construct Mix | Sigma | Catalog # SHP001 | |
| Chemical compound, drug | OPTI-MEM | Gibco | Catalog # 31985-062 | |
| Chemical compound, drug | Polybrene Transfection Reagent | Millipore | Catalog # TR-10030G | |
| Chemical compound, drug | Puromycin | InvivoGen | Catalog # Ant-pr | |
| Chemical compound, drug | Lipofectamine 3000 Transfection Reagent | Invitrogen | Catalog # L300075 | |
| Chemical compound, drug | Protein G Dynabeads | Invitrogen | Catalog # 10004D | |
| Chemical compound, drug | TCEP Bond Breaker | Thermo Fisher | Catalog # 77720 | |
| Chemical compound, drug | Halt Protease and Phosphatase Inhibitor | Thermo Fisher | Catalog # 78436 | |
| Chemical compound, drug | Benzonase | Sigma-Aldrich | Catalog # 70664-10KUN | |
| Chemical compound, drug | Dithiothreitol (DTT) | Sigma-Aldrich | Catalog # D0632-10G | |
| Chemical compound, drug | Iodoacetamide (IAA) | Sigma-Aldrich | Catalog # I1149-5G | |
| Chemical compound, drug | Lysyl Endopeptidase (LysC) | Fujifilm Wako Chemicals USA | Catalog # 125-05061 | |
| Chemical compound, drug | Formic acid | Fisher Chemical | Catalog # A117-50 | |
| Chemical compound, drug | Sep-Pak C-18 | Waters | Catalog # WAT036925 | |
| Chemical compound, drug | Trypsin | Promega | Catalog # V5111 | |
| Chemical compound, drug | Easy-Spray 50 cm column packed with 2 mm C-18 Resin | Thermo Fisher | Catalog # ES903 | |
| Chemical compound, drug | DNA Degradase Plus | Zymo Research | Catalog # 214843 | |
| Chemical compound, drug | 10× DNA Degrader Reaction Buffer | Zymo Research | Catalog # E2016-2 | |
| Chemical compound, drug | 2′-deoxyadenosine (dA) | Sigma | Catalog # D7400 | |
| Chemical compound, drug | N6-methyl-2-deoxyadenosine (6mA) | Thermo Fisher | Catalog #AAJ64961MD | |
| Chemical compound, drug | Stable heavy labeled 2′-deoxyadenosine | Cambridge Isotope Laboratories, Inc | CNLM-3896-CA-25; internal standard for analyte mass spectrometry | |
| Chemical compound, drug | NuPAGE LDS Sample Buffer (4×) | Invitrogen | Catalog # NP0007 | |
| Chemical compound, drug | NuPAGE Sample Reducing Agent (10×) | Invitrogen | Catalog # NP0009 | |
| Chemical compound, drug | MOPS SDS Running Buffer (20×) | Invitrogen | Catalog # NP0001 | |
| Chemical compound, drug | Invitrogen iBlot 2 Transfer Stacks, PVDF, mini | Invitrogen | Catalog # IB24002 | |

*Continued*

| Reagent type (species) or resource | Designation | Source or reference | Identifiers | Additional information |
|---|---|---|---|---|
| Chemical compound, drug | Chameleon Duo Prestained Protein ladder | LiCor | Catalog # 928-60000 | |
| Chemical compound, drug | Tuberculin Needle | BD | SKU: 309623 | |
| Chemical compound, drug | Intercept Blocking Buffer | LiCor | Catalog # 927-70001 | |
| Chemical compound, drug | Phosphate Buffered Saline-Tween (20×) | Boston Bioproducts Inc | Catalog # IBB-920 | |
| Chemical compound, drug | FBS | Gibco | Catalog # 16000-044 | |
| Chemical compound, drug | 1 M HEPES | Corning | Catalog # 25-060-CI | |
| Chemical compound, drug | 0.5 M EDTA | Invitrogen | Catalog # 46-000-CM | |
| Chemical compound, drug | NaCl | Sigma | Catalog # S3014-1K | |
| Chemical compound, drug | Triton X-100 | Thermo Scientific | Catalog # A16046.AE | |
| Chemical compound, drug | Sucrose | Thermo | Catalog # 036508.30 | |
| Chemical compound, drug | $MgCl_2$ | Fluka | Catalog # 63020-1L | |
| Chemical compound, drug | 37% Formaldehyde | Thermo Scientific | Catalog # BP531-25 | |
| Chemical compound, drug | DAPI | Thermo Scientific | Catalog # 62248 (use at 1:10,000) | |
| Chemical compound, drug | Duplex Buffer | IDT | Catalog # 1072570 | |
| Chemical compound, drug | Electroporation Enhancer | IDT | Catalog # 1075916 | |
| Chemical compound, drug | Alt-R CRISPR Cas9 tracrRNA | IDT | Catalog # 1073190 | |
| Chemical compound, drug | Alt-R S.p. Cas9 Nuclease V3 | IDT | Catalog # 1081059 | |
| Chemical compound, drug | Duplex Buffer | IDT | Catalog # 11-01-03-01 | |
| Chemical compound, drug | Amaxa SE Cell Line Kit | Lonza | Catalog # V4SC-1096 | |
| Chemical compound, drug | ---Solution Box | In Kit | Catalog # PBC1-02250 | |
| Chemical compound, drug | ---SE solution | In Kit | Catalog # S-09637 | |
| Chemical compound, drug | ---Supplement Solution | In Kit | Catalog # S-09699 | |
| Chemical compound, drug | TaqMan Gene Expression Master Mix | Thermo Fisher | Catalog # 4369016 | |
| Chemical compound, drug | Taqman Assay – GAPDH | Thermo Fisher | Catalog # 4331182, Assay ID Hs99999905_m1 | |
| Chemical compound, drug | Taqman Assay – UNG | Thermo Fisher | Catalog # 4331182, Assay ID Hs01037093_m1, | |

*Continued on next page*

*Continued*

| Reagent type (species) or resource | Designation | Source or reference | Identifiers | Additional information |
|---|---|---|---|---|
| Chemical compound, drug | DMEM for SILAC | Thermo Fisher | Catalog # 88364 | |
| Chemical compound, drug | $13C_6$ L-Arginine-HCl | Thermo Fisher | Catalog # 88210 | |
| Chemical compound, drug | $13C_6$ L-Lysine-2HCl | Thermo Fisher | Catalog # 88209 | |
| chemical compound, drug | Biotin-phenol | LGC GENOMICS LLC | Catalog # 41994-02-9 | |
| Chemical compound, drug | Doxycycline | Sigma-Aldrich | Catalog # D9891-1G | |
| Chemical compound, drug | $H_2O_2$ | Sigma-Aldrich | Catalog # H1009 | |
| Chemical compound, drug | Sodium ascorbate | Sigma-Aldrich | Catalog # A7631 | |
| Chemical compound, drug | Trolox | Sigma-Aldrich | Catalog # 238813 | |
| Chemical compound, drug | Sodium azide | Sigma-Aldrich | Catalog # S2002 | |
| Chemical compound, drug | Phosphate-buffered saline (PBS) | Gibco | Catalog # 10010049 | |
| Chemical compound, drug | cOmplete protease inhibitor cocktail | Sigma | Catalog # 11873580001 | |
| Chemical compound, drug | Methanol | Sigma | Catalog # 1793307 | |
| Chemical compound, drug | Crystal Violet | Aqua Solutions | Catalog # C8126 | |
| Software, algorithm | Model-based Analysis of Genome-wide CRISPR-Cas9 Knockout (MAGecK) | Model-based Analysis of Genome-wide CRISPR-Cas9 Knockout (MAGecK) | MLE, RRID:SCR_025016 | |
| Software, algorithm | ImageJ | ImageJ | 1.47v, RRID:SCR_003070 | |
| Software, algorithm | PRISM | Graph Pad Software | Version 9, RRID:SCR_002798 | |
| Software, algorithm | Sciex OS: Autopeak | Sciex | 2.2.0 | |
| Software, algorithm | CellProfiler | CellProfiler | This paper | |
| Software, algorithm | MaxQuant | MaxQuant | 1.6.17.0, RRID:SCR_014485 | |
| Other, equipment | CX7 CellNightSight | Thermo Fisher | Immunofluorescence | Microscope for immunofluorescence |
| Other, equipment | UltraView Spinning Disk | PerkinElmer | Cry2 Imaging, RRID:SCR_020405 | Microscope for Cry2 imaging |
| Other, equipment | Incucyte | Sartorius | Cell Viability – Growth, RRID:SCR_019874 | Incubator for cell viability assays |
| Other, equipment | Odyssey CX7 | Li-cor | Immunoblotting | Imager for western blots |
| Other, equipment | Illumina Next-Seq | Illumina | Whole Genome Screen | Sequencer for Whole Genome Screen |
| Other, equipment | Illumina Mi-Seq | Illumina | Whole Genome Screen | Sequencer for Whole Genome Screen |
| Other, equipment | 4D Nucleofector | Lonza | KO line generation, RRID:SCR_023155 | Nucleofector for generating CRISPR Kos |
| Other, equipment | Envision 2104 Plate Reader | PerkinElmer | Cell Viability – MTS | Plate Reader for MTS assay |

*Continued on next page*

*Continued*

| Reagent type (species) or resource | Designation | Source or reference | Identifiers | Additional information |
|---|---|---|---|---|
| Other, equipment | nanoACQUITY UPLC System | Waters | Coimmunoprecipitation LC–MS/MS | Equipment for LC–MS/MS |
| Other, equipment | Orbitrap Fusion Lumos Tribrid Mass Spectrometer | Thermo Fisher | Coimmunoprecipitation LC–MS/MS | Equipment for LC–MS/MS |
| Other, equipment | ACQUITY UPLC M Class System | Waters | UPLC–MS/MS | Equipment for UPLC–MS/MS Equipment for LC–MS/MS |
| Other, equipment | Triple QuadTM 7500 System | Sciex | UPLC–MS/MS | Equipment for UPLC–MS/MS Equipment for LC–MS/MS |
| Other, equipment | Column: nanoEase m/z peptide BEH c18, 300A, 1.7 µm 300 µm × 100 mm | Waters | UPLC–MS/MS, PN186009264 | Equipment for UPLC–MS/MS Equipment for LC–MS/MS |
| Other, GEO | GSE282260 | GEO | GSE282260 | |

## Resource availability

### Lead contact
Further information and requests for resources and reagents should be directed to and will be fulfilled, whenever possible, by the lead contact.

## Materials availability
There may be licensing restrictions to the availability of engineered cell lines and plasmids generated in this study.

## Experimental model details

### Method details

### Cell lines
SW620-, DLD-1-, and DLD-1-derived cell lines were maintained in RPMI-1640 + 10% fetal bovine serum (FBS) + 1× penicillin–streptomycin. HT-29- and HT-29-derived lines were maintained in McCoy's 5A + 10% FBS + 1× penicillin–streptomycin. U2OS 2-6-3 cells were grown in full DMEM supplemented with 10% FBS, penicillin–streptomycin and GlutaMAX. All cells were maintained at 37°C with 5% $CO_2$ in a humidified sterile incubator. Cell lines routinely tested negative for mycoplasma contamination.

### Generation of stable cell lines
To establish DLD-1 UNG KO cell lines, $2 \times 10^5$ DLD-1 cells were nucleofected with ribonucleoprotein complexes targeting UNG (UNG gRNA_UNG_4) or HPRT (gRNA_HPRT) using the SE Cell Line 4D-Nucleofector X Kit (Lonza) and CM-150 program on the 4D-Nucleofector (Lonza). Ribonucleoprotein complexes consisting of 104 pmol Cas9 (IDT) and 120 pmol trRNA:crRNA (1:1) (IDT), 2.5 µl electroporation enhancer (IDT) were prepared in SE nucleofection buffer (IDT) to a final volume of 25 µl per sample as described in IDT's Alt-R CRISPR-Cas9 System protocol and transferred to each well of a nucleofector 8-well strip. Following nucleofection, 75 µl of the appropriate pre-warmed culture medium was added to each well of the nucleofector 8-well strip and 50 µl of mixture was transferred to a 96-well plate. 48–72 hr after nucleofection, half of the cells were collected to extract DNA to assay cutting efficiency and the remaining cells were seeded for single-cell cloning. Knockout in clonal lines was confirmed by western blot.

To generate UNG2 cDNA expressing cell lines, cDNAs were delivered by retroviral or lentiviral transduction after packaging in HEK 293T cells. $5 \times 10^6$ were plated the evening before transfection. DNA and viral packaging vectors were transfected into cells with TransIT-293 transfection reagent according to the manufacturer's protocol. The media was changed the next day and after 24 hr, supernatants were harvested and filtered (0.45 µM). Harvests were repeated every 12 hr for 2 days. Target DLD-1 UNG KO cells were infected with virus-containing supernatants supplemented with 4 µg/ml polybrene. Stably expressing cells were selected with puromycin (0.5–2 µg/ml). See the resource for the list of plasmids used.

U2OS 2-6-3 cells were co-transfected with a plasmid expressing the PiggyBac transposase and a PiggyBac donor plasmid containing the doxycycline-inducible fusion protein GFP-LacI-APEX2-UNG2_IDR

and puromycin resistance. Cells were incubated in Lipofectamine 3000 Transfection Reagent for 48 hr and allowed to recover for 24 hr post-transfection. Cells were selected by incubating in full DMEM supplemented with 1.5 µg/ml of puromycin for 5–7 days. Selection media was changed every other day. Two stable cell lines were made. One with GFP-LacI-APEX2-UNG2_IDR (LA-UNG2_IDR) stably integrated and control with GFP-LacI-APEX2-STOP (LA-STOP) stably integrated.

## CRISPR/Cas9 screening

HT-29 cells were transduced with pLenti7-EF1a-Cas9 and cells were selected with 2 mg/ml hygromycin-containing medium. Cas9 expression was confirmed by western blot. Cas9 gene editing efficiency was confirmed by next-generation sequencing and colony-forming assay. Cas9-expressing HT-29 cells were transduced with a lentiviral sgRNA library, and split into four pools. Each pool was transduced at a MOI of 0.3 and 1 µg/ml puromycin-containing medium was added the next day. The selection was continued until 4 days post-transduction, which was considered the initial time point, t0. At this point, the transduced cells were divided into two populations and split into technical triplicates. One population was untreated and 2.2 nM floxuridine was added to the other. Cells were grown with or without floxuridine until t11 or t15, subculturing every 3–4 days. Cell pellets were frozen at each time point for gDNA isolation. A library coverage of ≥500 cells/sgRNA was maintained throughout the screen. gDNA from cell pellets was isolated using QIAGEN Gentra Puregene kit and genome-integrated sgRNA sequences were amplified by PCR using KOD Hot Start Polymerases. i5 and i7 multiplexing barcodes (Ilumina) were added in a second round of PCR and final gel-purified products were sequenced on Illumina HiSeq2500 or NextSeq500 systems to determine sgRNA representation in each sample. Gene knockouts enriched at t11 or t15 as compared to t0 were identified using Model-based Analysis of Genome-wide CRISPR–Cas9 Knockout (MAGeCK) analysis.

## Coimmunoprecipitation and mass spectrometry

Coimmunoprecipitation: DLD-1 UNG KO and DLD-1 HPRT cells were grown in the absence or presence of 16 nM floxuridine for 72 hr. Cells were collected and lysed in 50 mM Tris-HCl pH 7.5, 150 mM NaCl, 1% Triton, 300 mM KCl, 10% glycerol, 1× Halt Protease and Phosphatase Inhibitor, 1 mM Bond-Breaker TCEP Solution and benzonase (IP buffer) for 30 min on ice, vortexing every 10 min. Debris were pelleted at 15,000 × $g$ for 15 min and 40 µl of protein G dynabeads slurry, coupled to 5 µg of the indicated antibody (either UNG or IgG), was added to the supernatant and rotated for 1 hr at 4°C. Samples were washed in ice-cold IP buffer 3 × 5 min each wash. Beads were then eluted with 90 µl of LDS buffer + 1× reducing agent.

Sample processing, data acquisition, and analysis: Eluates were subjected to chloroform/methanol protein precipitation for removal of salts and detergents. An aliquot of eluate was first mixed with 4 volumes of methanol, followed by 1 volume of chloroform, and 3 volumes of water. Samples were vortexed vigorously and centrifuged for 5 min at 14,000 × $g$. The top aqueous layer was then discarded and 4 volumes of methanol were added before centrifuging for 5 min at 14,000 × $g$. The supernatant was removed, and the remaining protein pellet was dried using a SpeedVac. Protein pellets were dissolved in 100 mM Tris pH 8.0, before reduction and alkylation with dithiothreitol and iodoacetamide, respectively. Proteins were digested with LysC overnight at room temperature and digested with trypsin for 12 hr at 37°C. Digestion was stopped with the addition of formic acid. Tryptic peptides were desalted with Sep-Pak C-18.

LC–MS/MS was performed using a Waters nanoACQUITY UPLC System coupled to an Orbitrap Fusion Lumos Tribrid Mass Spectrometer. Peptide separation was carried out with an Easy-Spray 50 cm column prepacked with 2 µm C-18 resin. Mass spectrometric analyses were carried out in positive ESI automatically switching between survey (MS) and fragmentation (MS/MS) modes for the top 10 highest peaks. Both survey and fragmentation spectral scans were acquired in the Orbitrap analyzer, with resolution preset at $R$ = 60,000 and $R$ = 15,000, respectively. The most intense spectral peaks with assigned charge states of ≥2 were fragmented by higher energy collision dissociation at a threshold of 30 NCE. The isolation window was set at 0.5 $m/z$ while the dynamic exclusion was set at 90 s. Automatic Gain Control target was set for 1.2 × 105 ions with a maximum injection time of 100 ms.

Data processing protein identification and quantitation were performed using MaxQuant. Peptide identification was carried out using the Andromeda search engine by querying the Uniprot human

FASTA. LysC and trypsin were selected as digestion enzymes. Variable modifications were acetyl (protein-N-term), oxidation (M) and deamination (NQ) while carbamidomethyl (C) was selected as a fixed modification. The mass deviation threshold for the first search and main searches was 20 ppm, respectively. False discovery rate was set at 0.05 for both peptide-spectrum match and protein.

### Cell viability
Cells were plated in 96-well plates (Corning, 3585) at 10,000 cells per well and were subjected to a 9-point dose response of floxuridine. Confluence was measured during floxuridine treatment on an Incucyte instrument (Sartorius, Incucyte S3). At the endpoint, when untreated wells reached confluence, dishes were removed from the Incucyte and the MTS assay (Abcam, ab197010) was performed for a measure of cell viability. Cells were seeded and treated the same day as nucleofection for MTS assays assessing the nucleofected DLD-1 cells.

### Immunofluorescence
Cells were seeded at 5000 or 10,000 cells per well and subjected to a 60-hr treatment with indicated drugs in a 96-well glass-bottom imaging plate. Cells were fixed with 3.7% formaldehyde in PBS and permeabilized with 0.5% Triton X in PBS. For anti-N6-methyladenosine staining, prior to fixation, cells were pre-extracted for 5 min with a cytoskeletal extraction buffer (25 mM HEPES pH 7.5, 50 mM NaCl, 1 mM EDTA, 3 mM $MgCl_2$, 300 mM sucrose, 0.5% Triton X) to remove the cytoplasmic signal. Cells were blocked for 1 hr at room temperature in 5% FBS in PBS and stained overnight at 4°C with 60 µl of indicated antibodies per well. On the subsequent day, the primary antibody mixtures were aspirated, and wells were washed with 5% FBS in PBS three times before incubation with secondary antibody for 1 hr at room temperature. After incubation, the secondary antibody mixture was aspirated and cells were washed in PBS three times with DAPI in the second wash. Cells were left in 100 µl PBS and imaged on a CX7 CellNightSight (Thermo Fisher). Images were analyzed using CellProfiler using DAPI staining to create cell nucleic masks, and foci were identified using per-nucleus adaptive thresholding. Arbitrary cut-offs of >5 or >10 foci per nucleus were used to quantify nucleic as positive for foci formation depending on experimental variation of intensity in image sets.

### Immunoblotting
Cells were harvested and counted upon collection. An equal number of cells were lysed by resuspension in an equal volume of hot 2× NuPAGE LDS buffer + 1× NuPAGE sample reducing agent. Samples were either sonicated or passed through a tuberculin needle 10 times. Subsequently, samples were boiled for 5 min at 95°C. Equal amounts of protein were separated by sodium dodecyl sulfate–polyacrylamide gel electrophoresis (SDS–PAGE) on precast 4–12% Bis-Tris gels or 10% Bis-Tris gels. 10 µl of Chameleon Duo Prestained Protein Ladder was loaded in the first well of the gel. Gels were transferred onto the polyvinylidene fluoride (PVDF) membrane using an iBlot 2 gel transfer device (Invitrogen). Membranes were blocked for 1 hr in Intercept blocking buffer and incubated in primary antibodies for 2 hr at room temperature or overnight at 4°C. Membranes were washed in PBST (3 × 10 min) before being incubated with fluorescently conjugated secondary antibodies for 1 hr at room temperature, membranes were washed again and visualized by the LiCor imaging system (Odyssey CLx). See key resources for the list of antibodies.

### PONDR analysis
Human UNG2 protein sequence (MIGQKTLYSFFSPSPARKRHAPSPEPAVQGTGVAGVPEESGDAAAIPA KKAPAGQEEPGTPPSSPLSAEQLDRIQRNKAAALLRLAARNVPVGFGESWKKHLSGEFGKPYFIKLMG FVAEERKHYTVYPPPHQVFTWTQMCDIKDVKVVILGQDPYHGPNQAHGLCFSVQRPVPPPPSLENIYK ELSTDIEDFVHPGHGDLSGWAKQGVLLLNAVLTVRAHQANSHKERGWEQFTDAVVSWLNQNSNGLVFL LWGSYAQKKGSAIDRKRHHVLQTAHPSPLSVYRGFFGCRHFSKTNELLQKSGKKPIDWKEL) was entered into PONDR at https://www.pondr.com and analyzed using the VSL2 predictor.

### optoDroplet Cry2 assay
DLD-1 UNG KO cells expressing Cry2 constructs were made according to methods for generation of UNG2 cDNA expressing cell lines. Cells were plated in chamber wells using DMEM media with no phenol red and imaged the following day on a UltraView spinning disk confocal using a ×60 oil

immersion objective. Cells were exposed to 5 s of blue light (488 nm), before capturing mCherry fluorescence (560 nm) and a single Z plane was captured. This was repeated continuously over the course of 5–10 min before capturing new fields.

## Biotinylation of UNG–IDR interaction partners in cells

SILAC cell culture: U2OS 2-6-3 LA-stop cell line was seeded in 6-well at 30% confluency in DMEM for SILAC supplemented with 50 mg $^{13}C_6$ L-Arginine-HCl, 50 mg $^{13}C_6$ L-Lysine-2HCl, 10% dialyzed FBS, penicillin–streptomycin and GlutaMAX. The cells were expanded in SILAC media for five passages and heavy labeling incorporation of 99.5% was verified by proteomic analysis.

APEX2 reaction: U2OS 2-6-3 cells were grown to 80% confluency and incubated with 500 µM biotin-phenol and 1 µg/ml doxycycline in full DMEM for 24 hr. Cells were treated for 1 min with 1 mM $H_2O_2$ to initiate biotinylating reaction and washed three times with Quencher solution (10 mM sodium ascorbate, 5 mM Trolox, 10 mM sodium azide in DPBS). Cells were washed once with PBS, and collected by trypsinization, and pelleted at 500 × g.

Nuclear extract preparation for proteomic samples: ~5 × 10^7 U2OS cell pellet was resuspended in 10 ml of CE buffer 20 mM HEPES-KOH pH 7.9, 10 mM KCl, 5 mM MgCl$_2$, 1 mM EDTA, 0.1% NP-40, 10 mM sodium ascorbate, 10 mM sodium azide, 1 mM DTT, and cOmplete protease inhibitor cocktail and incubated on ice for 5 min before pelleting at 500 × g for 5 min at 4°C. Nuclear pellet was resuspended in 1 ml of NE buffer (20 mM HEPES-KOH pH 7.9, 500 mM NaCl, 1.5 mM MgCl$_2$, 0.2 mM EDTA, 0.5% NP-40, 1 mM DTT, and cOmplete protease inhibitor cocktail) and placed on a rotator for 1.5 hr at 4°C. Samples were centrifuged at maximum speed at 4°C for ~35 min and supernatant was collected.

SILAC lysate preparation: Protein concentration was measured using Quibit Protein Assay kit and equal concentrations of heavy control lysate was added to experimental light lysate samples for a sample mix test of 20 µl to be analyzed by mass spectrometry for 1:1 heavy:light ratio optimization. The mix test samples were adjusted to reach a 1:1 ratio as needed before final mix and subsequent pulldown.

Biotinylated protein pulldown: Pierce Streptavidin Magnetic Beads were washed three times in 1 ml TBST and once in 1 ml NE buffer (20 mM HEPES-KOH pH 7.9, 500 mM NaCl, 1.5 mM MgCl$_2$, 0.2 mM EDTA, 0.5% NP-40, 1 mM DTT, and cOmplete protease inhibitor cocktail). DynaMag-2 magnetic rack was used to collect beads and resuspend in 110 µl of NE buffer. Pre-washed streptavidin beads were added to fixed 1:1 heavy:light lysate mix and incubated on a rotator overnight at 4°C. Streptavidin beads were collected, and the supernatant was removed. Beads were washed as described (**Hung et al., 2016**) twice with RIPA lysis buffer (50 mM Tris, 150 mM NaCl, 0.1% (wt/vol) SDS, 0.5% (wt/vol) sodium deoxycholate, and 1% (vol/vol) Triton X-100 in Millipore water, pH 7.5), once with 1 M KCl, once with 0.1 M Na$_2$CO$_3$, once with 2 M urea in 10 mM Tris-HCl pH 8.0, twice with RIPA lysis buffer; wash buffers were kept on ice throughout the procedure. Biotinylated proteins were eluted from the beads in 60 µl 3× Laemmli buffer (6×: 0.35 M Tris-HCl pH 6.8, 30% glycerol, 10% SDS, 20% beta-mercaptoethanol, and 0.04% bromophenol blue) supplemented with 2 mM biotin for 10 min at 95°C. Samples were vortexed briefly, placed on ice for 3 min, and spun down briefly to bring down condensation. Proteomic samples were submitted on the same day.

## Quantitative real-time polymerase chain reaction

Total RNA was extracted from cells post-treatment with 60 hr of indicated treatment using RNeasy Plus Universal Kit according to the manufacturer's specifications. 5 µg of total RNA was reverse transcribed using Applied Biosystems High-Capacity cDNA Reverse Transcription Kit. The relative levels of genes of interest were determined by RT-qPCR using TaqMan Gene expression kit using 100 ng of cDNA in qPCR reactions. For qPCR reactions, TaqMan Gene expression Master Mix was prepared with GAPDH or UNG Taqman Assays in triplicate. Reactions were run and analyzed on an Applied Biosystems Quant Studio Flex 7 RT system.

## Biolayer interferometry

Biolayer interferometry binding assay for determination of kinetics and affinity of UNG protein to various single- and double-stranded DNA templates was performed on an Octet Red384 instrument (ForteBio, Inc). Binding experiments were carried out at 25°C in a binding buffer containing 25 mM Tris pH 7.5, and 10 mM NaCl.

Biotin-tagged DNA templates were captured on streptavidin-coated sensors (ForteBio, 18-5019). The capture step was carried out in a binding buffer. The plate shake speed was maintained at 1000 rpm throughout the experiment. After an initial baseline equilibration of 120 s, streptavidin-coated sensors were dipped in 3 µg/ml solution of biotin-tagged DNA template for 180 s to achieve capture levels of 0.2 nm. The sensors were dipped in buffer for 120 s to collect baseline signal before they were dipped in 150, 50, or 16.7 nM UNG protein in binding buffer for 300 s of association phase. The sensors were then immersed in the binding buffer for measuring 600 s of the dissociation phase. To account for any nonspecific binding, the signal for a sample well containing only binding buffer was used as blank and subtracted from all binding data. Binding curves were globally fit to a 1:1 Langmuir binding model using ForteBio Data Analysis 10.0 software to determine binding affinities, $KD$, from the kinetics data.

### DNA analyte detection by UPLC–MS/MS analysis

DLD-1 cells were seeded and treated with DMSO, 500 nM floxuridine, or 500 nM raltitrexed for 72 hr. Cells were washed, collected, and DNA was prepped using Quick-DNA/RNA Miniprep Plus Kit (Zymo Research). 1 µg of each sample was digested with 10 U of DNA degradase plus enzyme mix in 50 µl of 1× reaction buffer for 2 hr at 37°C. After incubation, samples were diluted with 50 µl molecular-grade water.

Separations were carried out on an ACQUITY UPLC M-Class System (Waters) with a nanoEase *m/z* peptide column (Waters) using a strong wash of 50:50 methanol:water and weak wash of 0.1% formic acid in water. The column temperature was maintained at 45°C. The mobile phases for the separations were 0.1% formic acid in water (A) and 0.1% formic acid in methanol (B). Initial conditions for the analysis were 98% A and decreased linearly for 6 min to 65% at a flow rate of 6.5 µl auth/min. All analytes eluted during this time. Samples were maintained at 10°C in the autosampler. Overall analysis time was 10.5 min which includes a wash step and column re-equilibration. The eluant was analyzed on a Triple Quad 7500 (Sciex) mass spectrometer. Due to the significant differences between the levels of analytes, separate analyses were performed for m6A versus dA. For additional mass spectrometry information, please see *Supplementary file 4*.

Analyte curves were created from stock solutions in water and serially diluted. A trendline was selected to offer the best fit and offer the widest analytical range. IS was added just prior to analysis to a final concentration of 1 nM for dA and 0.05 nM for m6A. For dA analysis, DNA samples were diluted 1:1000 in water prior to analysis and for m6A, the samples were diluted 1:5 in water.

### Mutational frequency assay

SW620 cells were cultured in HAT-containing medium to maintain a functional HPRT locus. Upon seeding cells for treatments with a METTL3 inhibitor, HAT-containing medium was replaced with a fresh medium. Cells were seeded at $0.5 \times 10^6$ cells per flask. Cells were grown for 7–9 days in the presence or absence of an METTL3 inhibitor. After the METTL3 inhibitor was removed, cells were re-seeded at $5 \times 10^2$ or $5 \times 10^4$ cells per 10 cm dish in triplicate. Dishes with $5 \times 10^4$ cells were treated with 5 µM 6-TG or medium-only control. Cells were grown for 12–16 days until visible colonies appeared. Subsequently, media on dishes was aspirated and cells were fixed in 1:1 methanol:crystal violet for 3 min at room temperature. Dishes were washed three times with 5 ml PBS. Images of dishes were acquired and the percentage of area containing particles was obtained using ImageJ.

## Quantification and statistical analysis

Image quantification was performed with CellProfiler. Statistical analysis was performed using Prism GraphPad. Statistical tests are referenced in figure legends.

## Acknowledgements

We thank Veronica Jové and Veronique Frattini for their thoughtful comments on the manuscript, and Pfizer's Postdoctoral Program for their support. We thank former Pfizer colleagues, Sarah Du, Chunying Zhao, and Alison Varghese for their help with generating reagents. We would also like to thank Pfizer colleagues in Emerging Science and Innovation, especially Paul Wes and Benedikt Bosbach, for stimulating questions, discussions, and technical expertise. This work was supported by Pfizer Inc.

# Additional information

## Competing interests

Brooke A Conti, Qing Xiang, Lei Shang, David J Shields, Mariano Oppikofer: previous employee of Pfizer and owns Pfizer stock. Leo Novikov, Deyan Tong: previous employee of Pfizer. Savon Vigil, Thomas J McLellan, Parag Sahasrabudhe, Jason D Arroyo: current employee of Pfizer and owns Pfizer stock. Chuong Nguyen: current employee of Pfizer. Reshma T Veettil, Prashant Pradhan, Benjamin R Sabari: P.P., R.T.V., N.D.L.C., and B.R.S., receivedresearch funding from Pfizer Inc. The other author declares that no competing interests exist.

## Funding

| Funder | Grant reference number | Author |
|--------|------------------------|--------|
| Pfizer's Centers for Therapeutic Innovation | | Savon Vigil |

The funders had no role in study design, data collection and interpretation, or the decision to submit the work for publication.

## Author contributions

Brooke A Conti, Conceptualization, Resources, Software, Formal analysis, Supervision, Funding acquisition, Validation, Investigation, Visualization, Methodology, Writing – original draft, Project administration, Writing – review and editing, Methodology: cell line generation, CRISPR screen, coimmunoprecipitations, cell viability, immunofluorescenceInvestigation: CRISPR screen, immunofluorescenceFormal Analysis: cell viability, immunofluorescence; Leo Novikov, Formal analysis, Investigation, Writing – review and editing, and Validation: cell viability, Visualization; Deyan Tong, Investigation, Writing – review and editing, Investigation: CRISPR Screen; Qing Xiang, Software, Formal analysis, Supervision, Methodology, Writing – review and editing, Methodology and Formal Analysis: CRISPR Screen; Savon Vigil, Formal analysis, Investigation, Methodology, Writing – review and editing, Investigation and Formal Analysis: UPLC-MS/MS for analytes; Thomas J McLellan, Resources, Formal analysis, Supervision, Investigation, Methodology, Writing – review and editing, UPLC-MS/MS for analytes; Chuong Nguyen, Formal analysis, Investigation, Writing – review and editing, Investigation and Formal Analysis: LC-MS/MS for coimmunoprecipitations; Nancy De La Cruz, Investigation, Methodology, Writing – review and editing, Investigation and Methodology: proximity biotinylation; Reshma T Veettil, Methodology, Writing – review and editing, Formal Analysis: proximity biotinylation; Prashant Pradhan, Investigation, Methodology, Writing – review and editing, Investigation and Methodology: proximity biotinylation; Parag Sahasrabudhe, Investigation, Methodology, Writing – review and editing, Investigation and Methodology: biolayer interferometry; Jason D Arroyo, Supervision, Writing – review and editing, Supervision: CRISPR Screen; Lei Shang, Formal analysis, Writing – review and editing, Formal analysis: CRISPR Screen; Benjamin R Sabari, Conceptualization, Supervision, Methodology, Project administration, Writing – review and editing, Supervision: proximity biotinylation; David J Shields, Resources, Supervision, Project administration, Writing – review and editing; Mariano Oppikofer, Conceptualization, Resources, Supervision, Funding acquisition, Visualization, Methodology, Writing – original draft, Project administration, Writing – review and editing

## Author ORCIDs

Brooke A Conti (ID) https://orcid.org/0000-0002-0320-0732
Mariano Oppikofer (ID) https://orcid.org/0000-0001-8579-0475

Reviewer #1 (Public review): https://doi.org/10.7554/eLife.101626.3.sa1
Reviewer #2 (Public review): https://doi.org/10.7554/eLife.101626.3.sa2
Author response https://doi.org/10.7554/eLife.101626.3.sa3

## Additional files

### Supplementary files

Supplementary file 1. CRISPR screen MAgeCK results, related to *Figure 1*.

Supplementary file 2. Liquid chromatograph–mass spectrometry (LC–MS/MS) analysis of coimmunoprecipitation using UNG antibody.

Supplementary file 3. Stable isotope labeling by amino acids (SILAC) liquid chromatograph–mass spectrometry (LC–MS/MS) analysis of IDR-seeded condensates, related to *Figure 3H*.

Supplementary file 4. Supplemental method information and vector sequences.

MDAR checklist

### Data availability

Whole-genome sequencing data have been deposited at GEO under accession code GSE282260. Accession numbers are listed in the key resources table. CellProfiler image analysis pipelines will be shared by the lead contact upon request. The paper does not report additional original code. Any additional information required to reanalyze the data reported in this paper is available from the lead contact upon request.

The following dataset was generated:

| Author(s) | Year | Dataset title | Dataset URL | Database and Identifier |
|---|---|---|---|---|
| Brooke AC, Deyan T, Qing X, Jason DA, Lei S | 2024 | N6-methyladenosine in DNA promotes genome stability | https://www.ncbi.nlm.nih.gov/geo/query/acc.cgi?acc=GSE282260 | NCBI Gene Expression Omnibus, GSE282260 |

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
