## [Editor Report · eLife Assessment]

This manuscript reports **important** findings that the methyltransferase METTL3 is involved in the repair of abasic sites and uracil in DNA, mediating resistance to floxuridine-driven cytotoxicity. The presented evidence is conclusive for the involvement of m6A in DNA involving single cell imaging and mass spectrometry data. The authors present **convincing** evidence that the m6A signal does not result from bacterial contamination or RNA.

---

## [Referee Report · Reviewer #1 (Public review)]

Summary:

The authors sought to identify unknown factors involved in the repair of uracil in DNA through a CRISPR knockout screen.

Strengths:

The screen identified both known and unknown proteins involved in DNA repair resulting from uracil or modified uracil base incorporation into DNA. The conclusion is that the protein activity of METTL3, which converts A nucleotides to 6mA nucleotides, plays a role in the DNA damage/repair response. The importance of METTL3 in DNA repair, and its colocalization with a known DNA repair enzyme, UNG2, is well characterized.

---

## [Referee Report · Reviewer #2 (Public review)]

Summary:

In this work, the authors performed a CRISPR knockout screen in the presence of floxuridine, a chemotherapeutic agent that incorporates uracil and fluoro-uracil into DNA, and identified unexpected factors, such as the RNA m6A methyltransferase METTL3, as required to overcome floxuridine-driven cytotoxicity in mammalian cells. Interestingly, the observed N6-methyladenosine was embedded in DNA, which has been reported as DNA 6mA in mammalian genomes and is currently confirmed with mass spectrometry in this model. Therefore, this work consolidated the functional role of mammalian genomic DNA 6mA, and supported with solid evidence to uncover the METTL3-6mA-UNG2 axis in response to DNA base damage.

Strengths:

In this work, the authors took an unbiased, genome-wide CRISPR approach to identify novel factors involved in uracil repair with potential clinical interest.

The authors designed elegant experiments to confirm the METTL3 works through genomic DNA, adding the methylation into DNA (6mA) but not the RNA (m6A), in this base damage repair context. The authors employ different enzymes, such as RNase A, RNase H, DNase, and liquid chromatography coupled to tandem mass spectrometry to validate that METTL3 deposits 6mA in DNA in response to agents that increase genomic uracil.

They also have the Mettl3-KO and the METTL3 inhibition results to support their conclusion.

Weaknesses:

The authors used the METTL3 inhibitor and Mettl3-KO to validate the METTL3-6mA-UNG2 functional roles. While not an outright weakness, rescue experiments of the KO line with wild type and the METTL3 catalytic mutant would have further strengthened the evidence.

---

## [Author Response]

The following is the authors’ response to the original reviews.

**eLife Assessment**
This manuscript reports important findings that the methyltransferase METTL3 is involved in the repair of abasic sites and uracil in DNA, mediating resistance to floxuridine-driven cytotoxicity. Convincing evidence shows the involvement of m6A in DNA based on single cell imaging and mass spec data. The authors present evidence that the m6A signal does not result from bacterial contamination or RNA, but the text does not make this overly clear.

We thank the editors for recognizing the importance of our work and the relevance of METTL3 and 6mA in DNA repair. We agree the evidence presented can be regarded as convincing, in that it includes validation with orthogonal approaches and excludes the source of 6mA being RNA or bacterial contamination.

To clarify, the identification of 6mA in DNA, upon DNA damage, is based first on immunofluorescence observations using an anti-m6A antibody. In this setting, removal of RNA with RNase treatment fails to reduce the 6mA signal, excluding the possibility that the source of signal is RNA. In contrast, removal of DNA with DNase treatment removes all 6mA signal, strongly suggesting that the species carrying the N6-methyladenosine modification is DNA (Figure 3D, E). Importantly, in Figure 3F, G, we provide orthogonal, quantitative mass spectrometry data that independently confirm this finding. Mass spectrometry-liquid chromatography of DNA analytes, conclusively shows the presence of 6mA in DNA upon treatment with DNA damaging agents and excludes that the source is RNA, based on exact mass.

Cells only show the 6mA signal when treated with DNA damaging agents, and the 6mA is absent from untreated cells (Figure 3D, E, H, I). This provides strong evidence that the 6mA signal is not a result of bacterial contamination in our cell lines. Furthermore, our cell lines are routinely tested for mycoplasma contamination. It could be possible that stock solutions of DNA damaging agents may be contaminated, but this would need to be true for all individual drugs and stocks tested, which is highly unlikely. Moreover, the data showing 6mA signal is not significantly different from untreated cells when a DNA damaging agent is combined with a METTL3 inhibitor (Figure 3H, I) provides strong evidence against bacterial contamination in our stocks.

In summary, we provide conclusive evidence, based on orthogonal methods, that the METTL3-dependent N6-methyladenosine modification is deposited in DNA, not RNA, in response to DNA damage and have now clarified these points in the results and discussion.

**Public Reviews:**

**Reviewer #1 (Public review):**
Summary:The authors sought to identify unknown factors involved in the repair of uracil in DNA through a CRISPR knockout screen.Strengths:The screen identified both known and unknown proteins involved in DNA repair resulting from uracil or modified uracil base incorporation into DNA. The conclusion is that the protein activity of METTL3, which converts A nucleotides to 6mA nucleotides, plays a role in the DNA damage/repair response. The importance of METTL3 in DNA repair, and its colocalization with a known DNA repair enzyme, UNG2, is well characterized.Weaknesses:This reviewer identified no major weaknesses in this study. The manuscript could be improved by tightening the text throughout, and more accurate and consistent word choice around the origin of U and 6mA in DNA. The dUTP nucleotide is misincorporated into DNA, and 6mA is formed by methylation of the A base present in DNA. Using words like 6mA "deposition in DNA" seems to imply it results from incorporation of a methylated dATP nucleotide during DNA synthesis.

The increased presence of 6mA during DNA damage could result from methylation at the A base itself (within DNA) or from incorporation of pre-modified 6mA during DNA synthesis. Our data do not directly discriminate between these two mechanisms, and we clarified this point in the discussion.

**Reviewer #2 (Public review):**
Summary:In this work, the authors performed a CRISPR knockout screen in the presence of floxuridine, a chemotherapeutic agent that incorporates uracil and fluoro-uracil into DNA, and identified unexpected factors, such as the RNA m6A methyltransferase METTL3, as required to overcome floxuridine-driven cytotoxicity in mammalian cells. Interestingly, the observed N6-methyladenosine was embedded in DNA, which has been reported as DNA 6mA in mammalian genomes and is currently confirmed with mass spectrometry in this model. Therefore, this work consolidated the functional role of mammalian genomic DNA 6mA, and supported with solid evidence to uncover the METTL3-6mA-UNG2 axis in response to DNA base damage.Strengths:In this work, the authors took an unbiased, genome-wide CRISPR approach to identify novel factors involved in uracil repair with potential clinical interest.The authors designed elegant experiments to confirm the METTL3 works through genomic DNA, adding the methylation into DNA (6mA) but not the RNA (m6A), in this base damage repair context. The authors employ different enzymes, such as RNase A, RNase H, DNase, and liquid chromatography coupled to tandem mass spectrometry to validate that METTL3 deposits 6mA in DNA in response to agents that increase genomic uracil.They also have the Mettl3-KO and the METTL3 inhibition results to support their conclusion.Weaknesses:Although this study demonstrates that METTL3-dependent 6mA deposition in DNA is functionally relevant to DNA damage repair in mammalian cells, there are still several concerns and issues that need to be improved to strengthen this research.First, in the whole paper, the authors never claim or mention the mammalian cell lines contamination testing result, which is the fundamental assay that has to be done for the mammalian cell lines DNA 6mA study.

Our cell lines are routinely tested for bacterial contamination, specifically mycoplasma, and we state this information in the revised manuscript.

Importantly, we do not observe 6mA in untreated cells, strongly suggesting that the 6mA signal observed is dependent on the presence of DNA damage and not caused by contamination in the cell lines (Figure 3D, E, H, I). While it could be possible that stock solutions of DNA damaging agents may be contaminated, this would need to be the case for all individual drugs and stocks tested that induce 6mA, which is very unlikely. Finally, the data showing 6mA signal is not significantly different from untreated cells when a DNA damaging agent is combined with a METTL3 inhibitor (Figure 3 H, I) provides strong evidence against bacterial contamination in our drug stocks.

Second, in the whole work, the authors have not supplied any genomic sequencing data to support their conclusions. Although the sequencing of DNA 6mA in mammalian models is challenging, recent breakthroughs in sequencing techniques, such as DR-Seq or NT/NAME-seq, have lowered the bar and improved a lot in the 6mA sequencing assay. Therefore, the authors should consider employing the sequencing methods to further confirm the functional role of 6mA in base repair.

While we agree that it could be important to understand the precise genomic location of 6mA in relation to DNA damage, this is outside the scope of the current study. Moreover, this exercise may prove unproductive. If 6mA is enriched in DNA at damage sites or as DNA is replicated, the genomic mapping of 6mA is likely to be stochastic. If stochastic, it would be impossible to obtain the read depth necessary to map 6mA accurately.

Third, the authors used the METTL3 inhibitor and Mettl3-KO to validate the METTL36mA-UNG2 functional roles. However, the catalytic mutant and rescue of Mettl3 may be the further experiments to confirm the conclusion.

We believe this to be an excellent suggestion from Reviewer #2 but we are unable to perform the proposed experiment at this time. We encourage future studies to explore the rescue experiment.

**Reviewer #3 (Public review):**
Summary:The authors are showing evidence that they claim establishes the controversial epigenetic mark, DNA 6mA, as promoting genome stability.Strengths:The identification of a poorly understood protein, METTL3, and its subsequent characterization in DDR is of high quality and interesting.Weaknesses:(1) The very presence of 6mA (DNA) in mammalian DNA is still highly controversial and numerous studies have been conclusively shown to have reported the presence of 6mA due to technical artifacts and bacterial contamination. Thus, to my knowledge there is no clear evidence for 6mA as an epigenetic mark in mammals, and consequently, no evidence of writers and readers of 6mA. None of this is mentioned in the introduction. Much of the introduction can be reduced, but a paragraph clearly stating the controversy and lack of evidence for 6mA in mammals needs to be added, otherwise, the reader is given an entirely distorted view of the field.These concerns must also be clearly in the limitations section and even in the results section which fails to nuance the authors' findings.

We agree with the reviewer that the presence and potential function of 6mA in mammalian DNA has been debated. Importantly, the debate regarding the presence and quantity of 6mA in DNA has been previously restricted to undamaged, baseline conditions. In complete agreement with this notion, we do not detect appreciable levels of 6mA in untreated cells. We revised the introduction section to present the debate about 6mA in DNA. We, however, want to highlight that our study provides, for the first time, convincing evidence (based on two orthogonal methods) that 6mA is present in DNA in response to a stimulus, DNA damage. We do not claim or provide any data that suggest 6mA is a baseline epigenetic mark.

(2) What is the motivation for using HT-29 cells? Moreover, the materials and methods do not state how the authors controlled for bacterial contamination, which has been the most common cause of erroneous 6mA signals to date. Did the authors routinely check for mycoplasma?

HT-29 is a cell line of colorectal origin and chemotherapeutic agents that introduce uracil and uracil derivatives in DNA, as those used in this study, are relevant for the treatment of colorectal cancer. As indicated above, we do not observe 6mA in untreated cells, strongly suggesting that the 6mA signal observed is dependent on DNA damage and not caused by a potential bacterial contamination (Figure 3D, E, H, I). Additionally, our cell lines are routinely tested for bacterial contamination, specifically mycoplasma.

(3) The single cell imaging of 6mA in various cells is nice. The results are confirmed by mass spec as an orthogonal approach. Another orthogonal and quantitative approach to assessing 6mA levels would be PacBio. Similarly, it is unclear why the authors have not performed dot-blots of 6mA for genomic DNA from the given cell lines.

We are confused by this point since an orthogonal approach to detect 6mA, mass spectrometry-liquid chromatography, was employed. This method does not use an antibody and confirms the increase of 6mA in DNA when cells were treated with DNA damaging agents. This data is presented in Figure 3F, G.

It is sensible to hypothesize that the localization of 6mA is consistent with DNA replication (like uracil deposition). In this event, the genomic mapping of 6mA is likely to be stochastic. This would make quantification with PacBio sequencing difficult because it would be very challenging to achieve the appropriate read depth to call a modified base.

Dot blots rely on an antibody and thus are not truly orthogonal to our immunofluorescence-based measurements. We preferred the mass spectrometry-liquid chromatography approach we took as a true orthogonal approach.

(4) The results of Figure 3 need further investigation and validation. If the results are correct the authors are suggesting that the majority of 6mA in their cell lines is present in the DNA, and not the RNA, which is completely contrary to every other study of 6mA in mammalian cells that I am aware of. This could suggest that the antibody is not, in fact, binding to 6mA, but to unmodified adenine, which would explain why the signal disappears after DNAse treatment. Indeed, binding of 6mA to unmethylated DNA is a commonly known problem with most 6mA antibodies and is well described elsewhere.

Based on this and the following comment, we are convinced that Reviewer #3 has overlooked two critical elements of our study:

First, the immunofluorescence work presented in Figure 3, showing 6mA signal in response to DNA damage, uses cells that were pre-extracted to remove excess cytoplasmic RNA. This method is often used in immunofluorescence experiments of this kind. The pre-extraction method removes most of the cytoplasmic content, and the majority of the cytoplasmic m6A RNA signal. Supplementary Figure 3D shows cells that have not been pre-extracted prior to staining. These images show the cytoplasmic m6A signal is abundant if we do not perform the pre-extraction step.

If the antibody used to label 6mA significantly reacted with unmodified adenine, we would expect a large signal in untreated or untreated and denatured conditions. In contrast, an increase in 6mA is not observed in either case.

Second, the orthogonal approach we employed, mass spectrometry coupled with liquid chromatography, measures 6mA DNA analytes specifically by exact mass. This approach does not depend on an antibody and yields results consistent with those from the immunofluorescence experiments.

(5) Given the lack of orthologous validation of the observed DNA 6mA and the lack of evidence supporting the presence of 6mA in mammalian DNA and consequently any functional role for 6mA in mammalian biology, the manuscript's conclusions need to be toned down significantly, and the inherent difficulty in assessing 6mA accurately in mammals acknowledged throughout.

As discussed in response to prior comments, Figure 3 does provide two independent and orthologous methods that demonstrate 6mA presence in DNA specifically, and not RNA, in response to DNA damage. Complementary and orthogonal datasets are presented using either immunofluorescence microscopy or mass spectrometry-liquid chromatography of extracted DNA. The latter method does not rely on an antibody and can discriminate 6mA DNA versus RNA based on exact mass. We revised the text to clarify that Figure 3F, G is a completely orthogonal approach.

**Recommendations for the authors:**

**Reviewer #2 (Recommendations for the authors):**
The authors cited most of the related publications; however, the reviewer suggested that three 2015 papers in Cell (Dahua Chen's, Yang Shi's, and Chuan He's) and the 2016 Nature (Andrew Xiao's) article are worth citing here because those are the milestone works reported the genomic DNA 6mA, for the first wave, in eukaryotic and mammalian genomes.Furthermore, in Tao P. Wu and Andrew Z. Xiao's 2016 Nature article, the result has already emphasized the genomic DNA 6mA is enriched in the H2A.X sites; therefore, that work indicated the link between DNA damage and repair and 6mA's functional role. The authors may add some comments or discussion on this point.Last but not least, the authors may also need to discuss the reported evidence of DNA 6mA's function in mitochondria.

We thank the reviewer for these suggestions. We revised our introduction and include additional references and discussion points, as suggested by the reviewer.

**Reviewer #3 (Recommendations for the authors):**
Minor points:(1) In general, the manuscript is too verbose, and the amount of text can be dramatically reduced/sharpened. The introduction in particular is too long.

We revised the manuscript and reduced text when appropriate.

(2) Each results section can also be condensed to improve clarity significantly. Indeed the results section reads like a 'Result & Discussion' section, which is then followed by a Discussion. Maybe the discussion section can be shortened to a 'conclusion'.

We revised the results section when appropriate and reworked the discussion.

Importantly, we revised the text related to Figure 3 as it does appear that Reviewer #3 did not appreciate key results present in this figure, specifically the orthogonal, mass spectrometry approach validating the discovery of 6mA DNA species (Figure 3F, G). We added a schematic as Figure 3F to further clarify this point as well.

(3) The accession number for sequencing data in GEO data should be provided.

The accession numbers is now provided in the manuscript. GSE282260.

(4) All figures are unnecessarily small and in some cases, supporting figures from the supplementary data should be moved into the main figure to improve clarity.

The figures are of high image quality and can be enlarged easily. If there are specific figures that the reviewer believes will improve clarity, we would be happy to move them.